# Prediction of type 2 diabetes mellitus onset using logistic regression-based scorecards

**Yochai Edlitz[1,2], Eran Segal[1,2]***

[1]Department of Computer Science and Applied Mathematics, Weizmann Institute of Science, Rehovot, Israel; [2]Department of Molecular Cell Biology, Weizmann Institute of Science, Rehovot, Israel

## Abstract

**Background:** Type 2 diabetes (T2D) accounts for ~90% of all cases of diabetes, resulting in an estimated 6.7 million deaths in 2021, according to the International Diabetes Federation. Early detection of patients with high risk of developing T2D can reduce the incidence of the disease through a change in lifestyle, diet, or medication. Since populations of lower socio-demographic status are more susceptible to T2D and might have limited resources or access to sophisticated computational resources, there is a need for accurate yet accessible prediction models.

**Methods:** In this study, we analyzed data from 44,709 nondiabetic UK Biobank participants aged 40–69, predicting the risk of T2D onset within a selected time frame (mean of 7.3 years with an SD of 2.3 years). We started with 798 features that we identified as potential predictors for T2D onset. We first analyzed the data using gradient boosting decision trees, survival analysis, and logistic regression methods. We devised one nonlaboratory model accessible to the general population and one more precise yet simple model that utilizes laboratory tests. We simplified both models to an accessible scorecard form, tested the models on normoglycemic and prediabetes subcohorts, and compared the results to the results of the general cohort. We established the nonlaboratory model using the following covariates: sex, age, weight, height, waist size, hip circumference, waist-to-hip ratio, and body mass index. For the laboratory model, we used age and sex together with four common blood tests: high-density lipoprotein (HDL), gamma-glutamyl transferase, glycated hemoglobin, and triglycerides. As an external validation dataset, we used the electronic medical record database of Clalit Health Services.

**Results:** The nonlaboratory scorecard model achieved an area under the receiver operating curve (auROC) of 0.81 (95% confidence interval [CI] 0.77–0.84) and an odds ratio (OR) between the upper and fifth prevalence deciles of 17.2 (95% CI 5–66). Using this model, we classified three risk groups, a group with 1% (0.8–1%), 5% (3–6%), and the third group with a 9% (7–12%) risk of developing T2D. We further analyzed the contribution of the laboratory-based model and devised a blood test model based on age, sex, and the four common blood tests noted above. In this scorecard model, we included age, sex, glycated hemoglobin (HbA1c%), gamma glutamyl-transferase, triglycerides, and HDL cholesterol. Using this model, we achieved an auROC of 0.87 (95% CI 0.85–0.90) and a deciles' OR of ×48 (95% CI 12–109). Using this model, we classified the cohort into four risk groups with the following risks: 0.5% (0.4–7%); 3% (2–4%); 10% (8–12%); and a high-risk group of 23% (10–37%) of developing T2D. When applying the blood tests model using the external validation cohort (Clalit), we achieved an auROC of 0.75 (95% CI 0.74–0.75). We analyzed several additional comprehensive models, which included genotyping data and other environmental factors. We found that these models did not provide cost-efficient benefits over the four blood test model. The commonly used German Diabetes Risk Score (GDRS) and Finnish Diabetes Risk Score (FINDRISC) models, trained using our data, achieved an auROC of 0.73 (0.69–0.76) and 0.66 (0.62–0.70),

*For correspondence:
eran.segal@weizmann.ac.il

**Competing interest:** The authors declare that no competing interests exist.

respectively, inferior to the results achieved by the four blood test model and by the anthropometry models.

**Conclusions:** The four blood test and anthropometric models outperformed the commonly used nonlaboratory models, the FINDRISC and the GDRS. We suggest that our models be used as tools for decision-makers to assess populations at elevated T2D risk and thus improve medical strategies. These models might also provide a personal catalyst for changing lifestyle, diet, or medication modifications to lower the risk of T2D onset.

**Funding:** The funders had no role in study design, data collection, interpretation, or the decision to submit the work for publication.

## Editor's evaluation

The authors have used the UK Biobank with sophisticated statistical modeling to predict the risk of type 2 diabetes mellitus development. Prognosis and early detection of diabetes are key factors in clinical practice, and the current data suggest a new machine-learning-based algorithm that further advances our ability to prevent diabetes.

## Introduction

Diabetes mellitus is a group of diseases characterized by symptoms of chronic hyperglycemia and is becoming one of the world's most challenging epidemics. The prevalence of type 2 diabetes (T2D) has increased from 4.7% in 1980 to 10% in 2021, and is considered the cause of an estimated 6.7 million deaths in 2021 (*International Diabetes Federation - Type 2 diabetes, 2022*). T2D is characterized by insulin resistance, resulting in hyperglycemia, and accounts for ~90% of all diabetes cases (*Zimmet et al., 2016*).

In recent years, the prevalence of diabetes has been rising more rapidly in low- and middle-income countries (LMICs) than in high-income countries (*Diabetes programme, WHO, 2021*). In 2019, Eberhard et al. estimated that every other person with diabetes in the world is undiagnosed (*Standl et al., 2019*). 83.8% of all cases of undiagnosed diabetes are in low-mid-income countries (*Beagley et al., 2014*), and according to the IDF Diabetes Atlas, over 75% of adults with diabetes live in low- to middle-income countries (*IDF Diabetes Atlas, 2022*), where laboratory diagnostic testing is limited (*Wilson et al., 2018*).

According to several studies, a healthy diet, regular physical activity, maintaining normal body weight, and avoiding tobacco use can prevent or delay T2D onset (*Home, 2022*; *Diabetes programme, WHO, 2021*; *Knowler et al., 2002*; *Lindström et al., 2006*; *Diabetes Prevention Program Research Group, 2015*). A screening tool that can identify individuals at risk will enable a lifestyle or medication intervention. Ideally, such a screening tool should be accurate, simple, and low-cost. It should also be easily available, making it accessible for populations having difficulties using the computer.

Several such tools are in use today (*Noble et al., 2011*; *Collins et al., 2011*; *Kengne et al., 2014*). The Finnish Diabetes Risk Score (FINDRISC), a commonly used, noninvasive T2D risk-score model, estimates the risk of patients between the ages of 35 and 64 of developing T2D within 10 years. The FINDRISC was created based on a prospective cohort of 4746 and 4615 individuals in Finland in 1987 and 1992, respectively. The FINDRISC model employs gender, age, body mass index (BMI), blood pressure medications, a history of high blood glucose, physical activity, daily consumption of fruits, berries, or vegetables, and family history of diabetes as the parameters for the model. The FINDRISC can be used as a scorecard model or a logistic regression (LR) model (*Bernabe-Ortiz et al., 2018*; *Lindström and Tuomilehto, 2003*; *Meijnikman et al., 2018*).

Another commonly used scorecard prediction model is the German Diabetes Risk Score (GDRS), which estimates the 5-year risk of developing T2D. The GDRS is based on 9729 men and 15,438 women between the ages of 35–65 from the European Prospective Investigation into Cancer and Nutrition (EPIC)-Potsdam study (*EPIC Centres - GERMANY, 2022*). The GDRS is a Cox regression model using age, height, waist circumference, the prevalence of hypertension (yes/no), smoking behavior, physical activity, moderate alcohol consumption, coffee consumption, intake of whole-grain bread, intake of red meat, and parent and sibling history of T2D (*Schulze et al., 2007*; *Mühlenbruch et al., 2014*).

**Table 1.** Cohort statistical data.

Characteristics of this study's cohort population and the UK Biobank (UKB) population. A '±' sign denotes the standard deviation. While type 2 diabetes (T2D) prevalence in the UKB participants is 4.8%, it is 1.79% in our cohort as we screened the cohort at baseline for HbA1c% levels <6.5%. The age range of the participants at the first visit was 40–69; thus, our models are not suitable for people who develop T2D at younger ages. The models predict the risk of developing T2D between the first visit to the UKB assessment center and the last visit. We refer to this feature as 'the time between visits'.

| | UKB population | Train, validation, and test sets | Test set | Train set | Validation set |
|---|---|---|---|---|---|
| Number of participants | 502,536 | 44,709 | 8,960 | 25,025 | 10,724 |
| Age at first visit (years) | 56.5 ± 8.1 | 55.6 ± 7.6 | 55.5 ± 7.5 | 55.6 ± 7.6 | 55.6 ± 7.6 |
| Age at last visit (years) | - | 62.9 ± 7.5 | 62.9 ± 7.4 | 62.9 ± 7.5 | 62.9 ± 7.5 |
| The time between visits (years) | - | 7.3 ± 2.3 | 7.3 ± 2.3 | 7.3 ± 2.3 | 7.3 ± 2.3 |
| Males in the population (%) | 45.5 | 47.8 | 47.9 | 47.9 | 47.5 |
| Diabetic at first visit (%) | 4.8 | 0 | 0 | 0 | 0 |
| Diabetic at last visit (%) | - | 1.79 | 1.76 | 1.75 | 1.91 |
| Hba1c at first visit (%) | 5.5 ± 0.6 | 5.3 ± 0.3 | 5.3 ± 0.3 | 5.3 ± 0.3 | 5.3 ± 0.3 |
| Hba1c at last return (%) | - | 5.4 ± 0.4 | 5.4 ± 0.3 | 5.4 ± 0.4 | 5.4 ± 0.4 |
| Weight at first visit (kg) | 78.1 ± 15.9 | 76.6 ± 14.7 | 76.4 ± 14.6 | 76.7 ± 14.7 | 76.8 ± 14.9 |
| Weight at last visit (kg) | - | 76.2 ± 15.2 | 76.0 ± 14.9 | 76.2 ± 15.2 | 76.5 ± 15.3 |
| Body mass index at first visit (kg/m²) | 27.4 ± 4.8 | 26.6 ± 4.2 | 26.5 ± 4.1 | 26.6 ± 4.2 | 26.7 ± 4.3 |
| Body mass index at last visit (kg/m²) | - | 26.6 ± 4.4 | 26.5 ± 4.3 | 26.6 ± 4.4 | 26.7 ± 4.5 |
| Hips circumference at first visit (cm) | 103.4 ± 9.2 | 102.1 ± 8.2 | 101.9 ± 8.0 | 102.1 ± 8.2 | 102.3 ± 8.3 |
| Hips circumference at last visit (cm) | - | 101.6 ± 8.8 | 101.4 ± 8.7 | 101.6 ± 8.8 | 101.8 ± 9.0 |
| Waist circumference at first visit (cm) | 90.3 ± 13.5 | 87.9 ± 12.5 | 87.7 ± 12.4 | 87.9 ± 12.4 | 88.2 ± 12.7 |
| Waist circumference at last visit (cm) | - | 88.7 ± 12.7 | 88.5 ± 12.5 | 88.7 ± 12.7 | 89.0 ± 12.9 |
| Height at first visit (cm) | 168.4 ± 9.3 | 169.5 ± 9.2 | 169.5 ± 9.1 | 169.5 ± 9.2 | 169.4 ± 9.1 |
| Height at last visit (cm) | - | 169.0 ± 9.2 | 169.0 ± 9.2 | 169.0 ± 9.3 | 168.9 ± 9.2 |

Barbara Di Camillo et al. reported in 2019 the development of three survival analysis models using the following features: background and anthropometric information, routine laboratory tests, and results from an Oral Glucose Challenge Test (OGTT). The cohorts consisted of 8483 people from three large Finnish and Spanish datasets. They report achieving area under the receiver operating curve (auROC) scores equal to 0.83, 0.87, and 0.90, outperforming the FINDRISC and Framingham scores (*Di Camillo et al., 2018*). In 2021, Lara Lama et al. reported using a random forest classifier on 7949 participants from the greater Stockholm area to investigate the key features for predicting prediabetes and T2D onset. They found that BMI, waist–hip ratio (WHR), age, systolic and diastolic blood pressure, and a family history of diabetes were the most significant predictive features for T2D and prediabetes (*Lama et al., 2021*).

The goal of the present research is to develop easy-to-use, clinically usable models that are highly predictive of T2D onset. We developed two simple scorecard models and compared their predictive power to the established FINDRISC and GDRS models. We trained both models using a subset of data from the UK Biobank (UKB) observational study cohort and reported the results using holdout data from the same study. We based one of the models on easily accessible anthropometric measures and the other on four common blood tests. Since we trained and evaluated our models using the UKB database, the models are therefore most relevant for the UK population aged 40–65 or for populations with similar characteristics (as presented in *Table 1*). As an external test case for the four blood test model, we used the Israeli electronic medical record database of Clalit Health Services (*Artzi et al., 2020*).

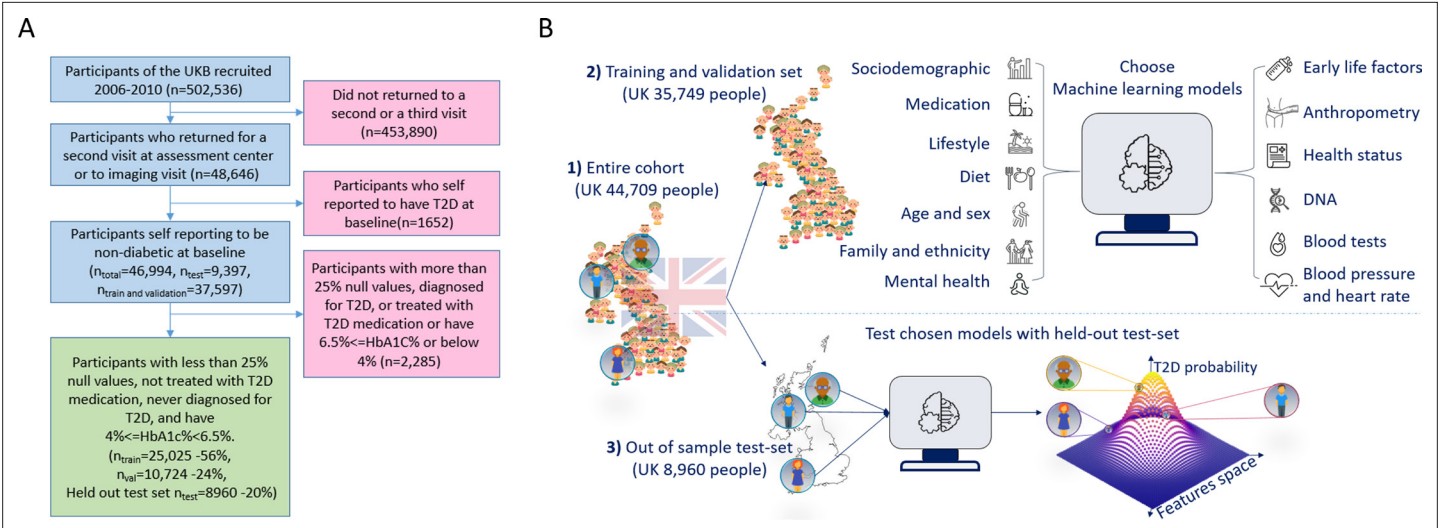

**Figure 1.** A flowchart of the cohort selection process and an illustrative figure of the model's extraction.
(**A**) A flowchart of the selection process of participants in this study. We selected participants who came for a repeated second or third visit from the 502,536 participants of the UK Biobank (UKB). Next, we excluded 1652 participants who self-reported having type 2 diabetes (T2D). We then split the data into 80% for the training and validation sets and 20% for the holdout test set. We excluded an additional 2285 participants due to (1) having 25% or more missing values from the full feature list, (2) having HbA1c levels above or equal to 6.5%, or (3) being treated with metformin or insulin, (4) found to be diagnosed with T2D before the first UKB visit. The final training, validation, and test sets included 25,025 participants (56% of the cohort), 10,724 participants (24%), and 8960 participants (20%), respectively. (**B**) The process flow during the training and testing of the models. We first split the data and kept a holdout test set. We then explored several models using the training and validation datasets. We then compared the selected models using the holdout test set and reported the results. We calibrated the output of the models to predict the probability of a participant developing T2D.

## Methods

### Data

We analyzed UKB's observational data of 502,536 participants aged 40–69 recruited in the UK voluntarily from 2006 to 2010. During a baseline assessment visit to the UKB, the participants self-completed questionnaires, which included lifestyle and other potentially health-related questions. The participants also underwent physical and biological measurements. Out of this cohort, we used the data of 20,346 participants who revisited the UKB assessment center from 2012 to 2013 for an additional medical assessment. We also used the data of 48,705 participants who revisited for a second or third visit from 2014 onward for an imaging visit and underwent an additional, similar medical check. We screened the participants to keep only those not being treated for nor having in the past T2D. We also screened out participants whose average blood sugar level for the past 2–3 months (hemoglobin A1c [HbA1c%]) was below 4% or above 6.5%. We started with 798 features for each participant and removed all the features with more than 50% missing data points in our cohort. We later screened out all the participants who still had more than 25% missing data points from the cohort and imputed the remaining missing data. We further removed those study participants who self-reported as being healthy but had HbA1c% levels higher than the accepted healthy level. We also screened participants who had a record of a prior T2D diagnosis (data field 2976 at the UKB). As not all participants had HbA1c% measurements, we estimated the bias of participants reporting as healthy while having an HbA1c% level indicating diabetes. For this estimate, we used the data from a subpopulation of our patients and found that 0.5% of participants reported being healthy with a median HbA1c% value of 6.7%, while the cutoff for having T2D is 6.5% (*Table 1*).

Of the remaining 44,709 participants in our study cohort, 1.79% developed T2D during a follow-up period of 7.3 ± 2.3 years (*Table 1*, *Figure 1A*). As a predicted outcome, we used the data for whether a participant develops T2D between the first and last visit from a self-report using a touchscreen questionnaire. The participants were asked to mark either 'Yes'/'No'/'Do not know'/'Prefer not to answer' for the validity of the sentence "Diabetes diagnosed by a doctor," which was presented to them on a touch screen questionnaire (data field 2443 at the UKB).

## Feature selection process

We started with 798 features that we hypothesized as potential predictors for T2D onset. We removed all the features with more than 50% missing data values, leaving 279 features for the research. Next, we imputed the missing data of the remaining records (see 'Methods'). As a genetic input for several models, we used both polygenic risk scores (PRS) and single-nucleotide-polymorphisms (SNPs) from the UKB SNP array (see 'Methods'). We used 41 PRSs with 129 ± 37.8 SNPs on average for each PRS. We also used the single SNPs of each PRS as some of the models' features; after removing duplicate SNPs, we remained with 2267 SNPs (see 'Genetic data').

We aggregated the features into 13 separate groups: age and sex; genetics; early-life factors; socio-demographic; mental health; blood pressure and heart rate; family history and ethnicity; medication; diet; lifestyle and physical activity; physical health; anthropometry; and blood test results. We trained models for each group of features separately (*Appendix 1—figure 1*, *Appendix 1—table 1*). We then added the features groups according to their marginal predictability (*Appendix 1—table 2*).

After selecting the leading models from the training and validation datasets, we tested and reported the results of the selected models using the holdout test set samples (*Appendix 1—table 1*). To encourage clinical use of our models, we optimized the number of features the models require. To simplify our models, we iteratively removed the least contributing features of our models using the training dataset (see 'Missing data', *Appendix 1—figure 1*). We examined the normalized coefficient of each model feature to assess its importance in the model. For the four blood test model, we initially also had 'reticulocytes' as one of the model's features. As we want to use common blood tests only, we dropped this feature from the list after confirming that the impact of removal of this feature on the model accuracy was negligible. Once the models were finalized, we developed corresponding scorecards that were both simple and interpretable (see 'Scorecards creation').

## Outcome

Our models provide a prediction score for the participant's risk of developing T2D during a specific time frame. The mean prediction time frame in our cohort is 7.3 ± 2.3 years. The results that we report correspond to a holdout test set comprising 20% of our cohort that we kept aside until the final report of the results. We also report the results of the four blood test model using an external electronic medical record database of the Israeli Clalit Health Services. We trained all the models using the same training set and then reported the test results of the holdout test set. We used the auROC and the average precision score (APS) as the main metrics of our models. Using these models, a physician can inform patients about their risk of developing T2D and their predicted risk of developing T2D within a selected time frame.

We calibrated the models to report the probability of developing T2D during a given time frame (see 'Calibration in methods'). To quantify the risk groups in the scorecards model, we performed a bootstrapping process on our validation dataset like the one performed for the calibration. We selected boundaries that showed good separation between risk groups and reported the results using the holdout test set.

## Missing data

After removing all features with more than 50% missing data and removing all the participants with more than 25% missing features, we imputed the remaining data. We analyzed the correlations between predictors with missing data and found correlations within anthropometry group features to other features in the same domain – analogous correlations were found in the blood test data. We used SKlearn's iterative imputer with a maximum of 10 iterations for the imputation and tolerance of 0.1 (*Abraham et al., 2014*) We imputed the training and validation sets apart from the imputation of the holdout test set. We did not perform imputation on the categorical features but transformed them into 'one hot encoding' vectors with a bin for missing data using Pandas categorical tools.

## Genetic data

We use PRS andSNPs as genetic input for some models. We calculated the PRS by summing the top correlated risk allele effect sizes derived from Genome-Wide Association Studies (GWAS) summary statistics. We first extracted from each summary statistics the top 1000 SNPs according to their p-value. We then used only the SNPs presented in the UKB SNP array. We used 41 PRSs with 129 ±

37.8 SNPs on average for each PRS. We also used the single SNPs of each PRS as features for some models. After the removal of duplicated SNPs, we kept 2267 SNPs as features. The full PRS summary statistics list can be found in Appendix 1, 'References for PRS summary statistics articles.' We calculated the PRS scores according to summary statistics publicly available from studies not derived from the KB to prevent data leakage.

## Baseline models

As the reference models for our results, we used the well-established FINDRISC and GDRS models (*Lindström and Tuomilehto, 2003*; *Schulze et al., 2007*; *Mühlenbruch et al., 2014*), which we retrained and tested on the same data used for our models. These two models are based on Finnish and German populations with similar age ranges as our cohort. We derived a FINDRISC score for each participant using the data for age, sex, BMI, waist circumference, and blood pressure medication provided by the UKB. To calculate the score for duration of physical activity, as required by the FINDRISC model, we summed up the values of 'duration of moderate activity' and 'duration of vigorous activity' as provided by the UKB. As a measure of the consumption of vegetables and fruits, we summed up the categories 'cooked vegetable intake,' 'salad/raw vegetable intake,' and 'fresh fruit intake' categories from the UKB. As an answer to the question '*Have any members of your patient's immediate family or other relatives been diagnosed with diabetes (type 1 or type 2)? This question applies to blood relatives only,*' we used the fields for the illness of the mother, the father, and the siblings of each participant.

We lacked the data of participants' grandparents, aunts, uncles, first cousins, and children. We also lacked the data about past blood pressure medication, although we do have the data for the current medication usage. Following the calculation of the FINDRISC score for each participant, we trained an LR model using the score for each participant as the model input and the probability of developing T2D as the output. We also examined an additional model, in which we added the time of the second visit as an input for the FINDRISC mode but found no major differences when this additional parameter was used. We report here the results for the FINDRISC model without the time of the second visit as a feature.

To derive the GDRS-based model, we built a Cox regression model using Python's lifelines package (*Davidson-Pilon et al., 2020*). As for the features of the GDRS model, we incorporated the following features: years between visits; height; prevalent hypertension; physical activity (hr/week); smoking habits (former smoker <20 units per day or ≥20 units per day, current smoker ≥20 units per day or <20 units per day); whole bread intake; coffee intake; red meat consumption; one parent with diabetes; both parents with diabetes; and a sibling with diabetes. We performed a random hyperparameter search the same way we used for our models. The hyperparameters we used here are the penalize parameter in the range of 0–10 using a 0.1 resolution and a variance threshold of 0–1 with 0.01 resolution. This last hyperparameter is used to drop columns where the variance of the column was lower than the variance threshold.

## Model building procedures

To test overfitting and biased models, we split the data into three groups: 20% for the holdout test set, used only for the final reporting of results. For the remaining data, we used 30% for the validation set and 70% for the training set. We use a two-stage process to evaluate the models' performance: exploration and test phases (*Figure 1*, *Appendix 1—figure 1*). We selected the optimal features during the exploration stage using the training and validation datasets. We then performed 200 iterations of a random hyperparameters selection process for each group of features. We set the selection criterion as the auROC metric using fivefold cross-validation.

We used the validation dataset to rank the various models by their auROC scores. We trained each of the models using the full training set with the top-ranked hyper-parameters determined from the hyperparameters tuning stage and ranked the models by their score using the validation dataset.

During the test phase, we reported the results of our selected models. For this, we evaluated the selected models using the holdout test set. To do so, we reran the hyperparameters selection process using the integrated training and validation datasets. We evaluated the trained model based on the data from the holdout test set. The same datasets were used for all the models.

For the development of the Cox regression models, we used the lifelines survival analysis package (*Davidson-Pilon et al., 2020*), using the 'age diabetes diagnosed' category (data field 2976) as a label. We used SKlearn's LogisticRegressionCV model for the LR model's computation (*Abraham et al., 2014*). For the Gradient Boosting Decision Trees (GBDT) models, we used Microsoft's LightGBM package (*Ke et al., 2017*). We developed our data pipeline to compute the scorecards. These last three models used the 'diabetes diagnosed by a doctor' category of the UKB as a label (data field 2443).

As part of the models' calculation process, we used 200 iterative random hyperparameters searches for the training of the models. For the GBDT models, we used the following parameter values for the searches: number of leaves – [2, 4, 8, 16, 32, 64, 128], number of boosting iterations – [50, 100, 250, 500, 1000, 2000, 4,000], learning rate – [0.005, 0.01, 0.05], minimum child samples – [5, 10, 25, 50], subsample – [0.5, 0.7, 0.9, 1], features fraction – [0.01, 0.05, 0.1, 0.25, 0.5, 0.7, 1], lambda l1 – [0, 0.25, 0.5, 0.9, 0.99, 0.999], lambda l2 – [0, 0.25, 0.5, 0.9, 0.99, 0.999], bagging frequency – [0, 1, 5], and bagging fraction – [0.5, 0.75, 1] (*Ke et al., 2017*).

We used a penalize in the range 0–2 with 0.02 resolution for the l2 penalty during the hyperparameters searches for the LR models.

We composed an anthropometric-based scorecard model to provide an accessible, simple, nonlaboratory, and noninvasive T2D prediction model. In this model, a patient can easily mark the result in each of the scorecard questions, consisting of the following eight parameters: age, sex, weight, height, hip circumference, waist circumference, BMI, and the WHR (*Figure 2A*).

In addition, we developed a more accurate tool for predicting T2D onset for those cases where laboratory testing will be available. We started with a feature selection process from a full-feature GBDT model, using only the training and validation datasets. We clustered the features of this model into 13 categories such as lifestyle, diet, and anthropometrics (*Appendix 1—table 1*, *Appendix 1—table 2*). We concluded that the blood tests have higher predictability than the other features aggregations. We thus trained a full blood test model using 59 blood tests available in the training dataset. Applying a recursive features' elimination process to the top 10 predictive features, we established the features of our final model based on four blood tests.

The four blood tests that we used are the glycated hemoglobin test (HbA1c%), which measures the average blood sugar for the past 2–3 months and which is one of the means to diagnose diabetes; gamma-glutamyl transferase test (GGT); high-density lipoprotein (HDL) cholesterol test, and the triglycerides test. We also included the time to prediction (time between visits); gender, age at the repeated visit; and a bias term related to the population's prevalence. We computed the values of these features' associated coefficients with their 95% CI to reconstruct the models (*Figure 3E*).

We tested the models in mixed and stratified populations of 1006 prediabetes participants with a T2D prevalence of 9.4% and a separate 7948 normoglycemic participants with a T2D onset prevalence of 0.8% (see Table 4).

## Shapley additive explanations (SHAP)

We used the SHAP method, which approximates Shapley values, for the feature importance analysis of the GBDT model. This method originated in game theory to explain the output of any machine-learning model. SHAP approximates the average marginal contributions of each model feature across all permutations of the other features in the same model (*Lundberg and Lee, 2017*).

## Predictors

To estimate the contribution of each feature's domain and for the initial screening of features, we built a GBDT model based on 279 features plus genetics data originating from the UKB SNPs array. We used T2D-related summary statistics from GWAS designed to find correlations between known genetic variants and a phenotype of interest. We used only GWASs from outside the UKB population to avoid data leakage (see supplementary material Appendix 1, 'References for PRS summary statistics articles').

We trained and tested the full-features model using the training and validation cohort to select the most predictive features for the anthropometry and the blood tests models. We then used this model's feature importance to extract the most predictive features. We omitted data concerning family relatives with T2D from the model as we did not see any major improvement over the anthropometrics

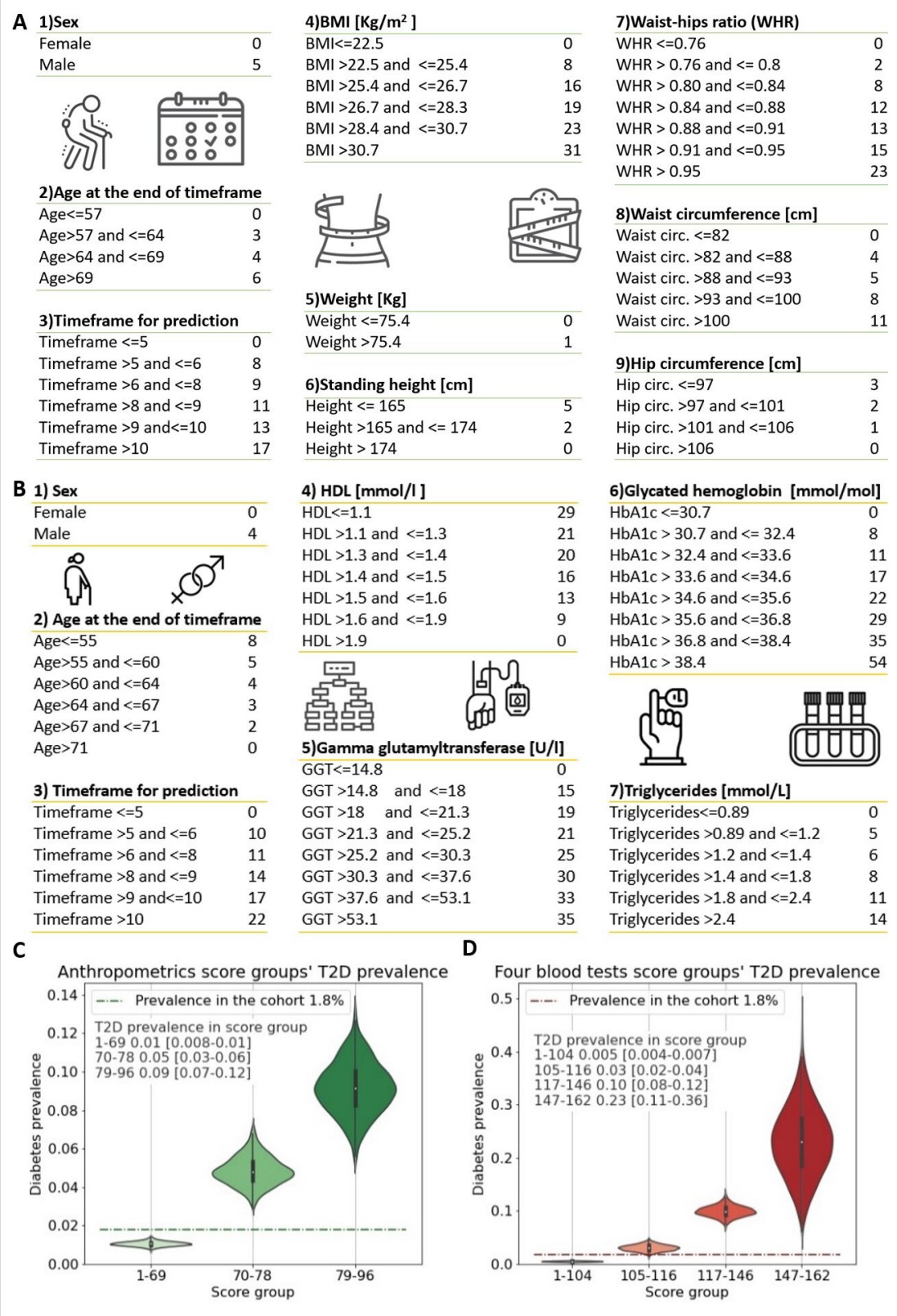

**Figure 2.** Anthropometrics and blood tests scorecards. (**A**) Anthropometrics-based scorecard. Summing the scores of the various features provides a final score that we quantified into one of three risk groups (figure 2C). (**B**) "Four blood test" scorecard. Adding the scores of the various features provides a final score that we quantified into one of four risk groups (Figure 2D). (**C**) Anthropometrics scorecards risk groups - first group score range [1-69] 1% [0.8-1%] 95%CI of developing T2D which is below the cohorts 1.8% prevalence of T2D (green dashed line); Second group, score range 70-78

*Figure 2 continued on next page*

*Figure 2 continued*

predicts a 5% [3-6%] 95%CI of developing T2D; Third group 79-96 9% [7-12%] 95%CI of developing T2D. (**D**) four blood tests scorecards risk groups - first group score range [1-104] <0.5% [0.04-0.7%] 95%CI of developing T2D which is below the cohorts 1.8% prevalence of T2D (red dashed line); Second group, score range 105-116 predicts a 3% [2-4%] 95%CI of developing T2D.; Third group range 117-146 with 10% [8-12%] 95%CI of developing T2D. Fourth group range 147-162 predicts 23% [10-37%] 95%CI of developing T2D, which is X13 fold prevalence enrichment compared to the cohort's T2D prevalence.

model. For the last step, we tested and reported the model predictions using data in the holdout section of the cohort.

For the extraction of the four blood test model, we performed a features selection process using the training and validation datasets. We executed models starting with 20 and down to 4 features of blood tests together with age and sex as features, each time removing the feature with the smallest importance score. We then selected the model with four blood tests (HbA1c%, GGT, triglycerides, HDL cholesterol) plus age and sex as the optimal balance between model simplicity (a small number of features) and model accuracy. We reported model results against data from the holdout test set.

We normalized all the continuous predictors using the standard 'z-score.' We normalized the train validation sets apart from the holdout test set to avoid data leakage.

## Model calibration

Calibration refers to the concurrence between the real T2D onset occurrence in a subpopulation and the predicted T2D onset probability in this population. Since our data are highly imbalanced between healthy and T2D ill patients, with a prevalence of 1.79% T2D, we used 1000 bootstrapping iterations of each model to improve the calibration. To do this, we first split each model's prediction into 10 deciles bins from 0 to 1 to calculate the calibration curves. Using Sklearn's isotonic regression calibration, we scale the results with a fivefold cross-validation (*Abraham et al., 2014*). We do so for each of the bootstrapping iterations. Lastly, we concatenate all the calibrated results and calculate each probability decile's overall mean predicted probability.

We split the probabilities range (0–1) into deciles (*Figure 3F*, *Figure 4*) and assigned each prediction sample to a decile bin according to the calibrated predicted probability of T2D onset.

## Scorecards creation

We used the training and validation datasets for our scorecards building process. We reported the results on the holdout dataset. We calculated our data's weight of evidence (WoE) by splitting each feature into bins. We binned greater importance features in a higher resolution while maintaining a monotonically increasing WoE (*Yap et al., 2011*). For quantizing the risk groups of the scorecards model, we performed 1000 iterations of a bootstrapping process on our validation dataset. We considered several potential risk score limits that separate T2D onset probability in each score group using the validation dataset. Once we set the final boundaries of the score, we reported the prevalence in each risk group on the test set. For the Cox regression-based scorecards, we used the parameters coefficient the same way we used the coefficient in the LR model for binning the model. When using a Cox regression-based scorecard, we compute the probability to develop based on a fix time frame for all participants (5 and 10 years' time frames models; Table 3). To enable an easy way for choosing the desired time frame as part of scorecard usage, we chose to use the LR-based scorecards as our model of choice for an additional development and validation.

## External validation cohort: EHR database of Clalit Health Services

As an external validation cohort for the four blood test scorecard model, we used the Clalit retrospective cohort's electronic health records. Clalit is the largest Israeli healthcare organization, serving more than 4.4 million patients (about half the population of Israel). The Clalit database holds electronic health records of over 11 million patients, dating back to 2002. It is considered one of the world's largest EHR databases (*Artzi et al., 2020*). We extracted data from patients who visited Clalit clinics from 2006 to 2011 and had a minimum of three HbA1C% tests, with the following inclusion criteria: the first sample below 6.5%, and two consecutive tests consistent with either HbA1c% ≥ 6.5 for each of the tests or both tests with HbA1C% < 6.5%. These were some of the criteria we used to indicate if the patient had developed T2D. We started with 179,000 patients that met the HbA1c%

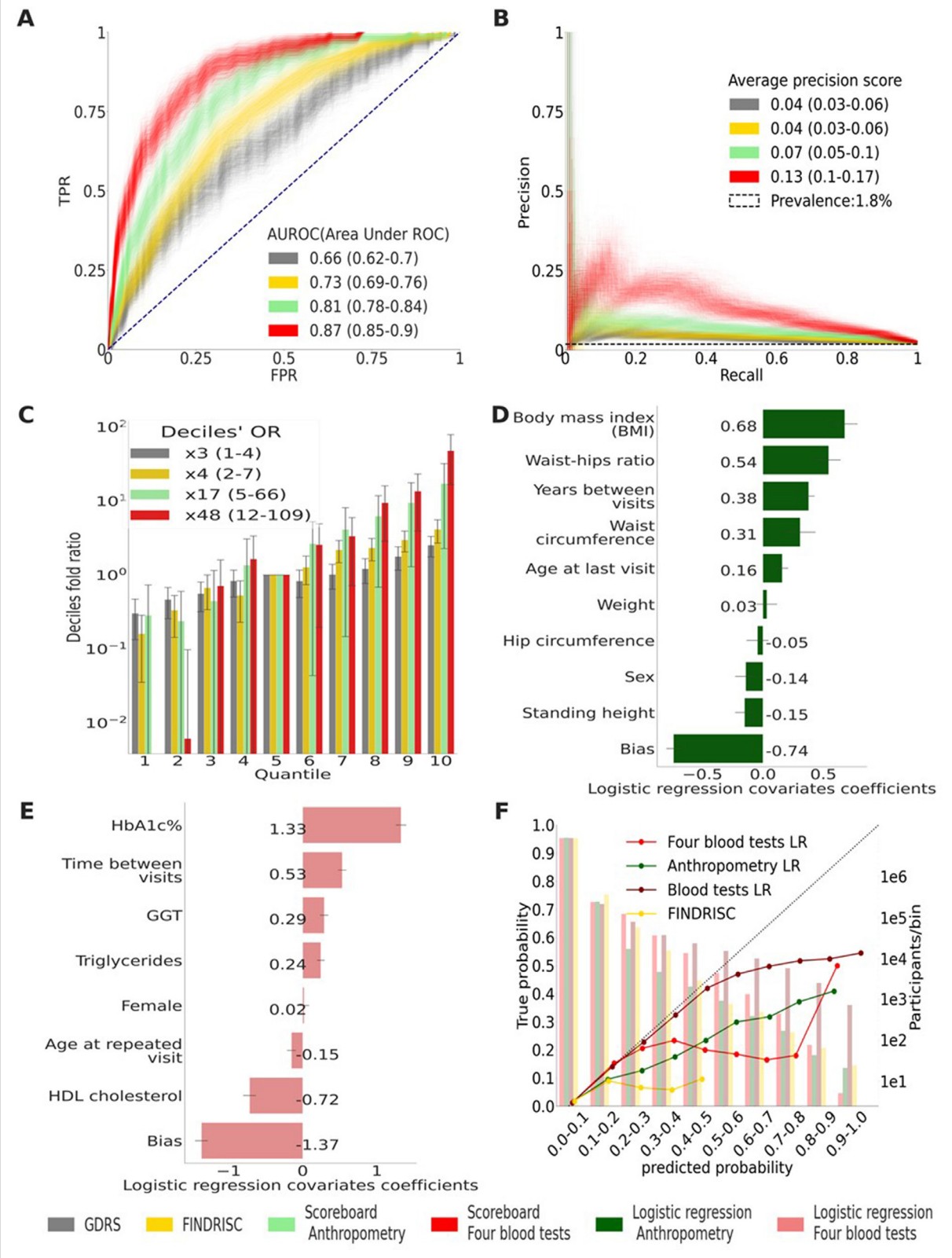

**Figure 3.** Main results calculated using 1000 bootstraps of the cohort population. Each point in the graphs represents a bootstrap iteration result. The color legend is at the bottom of the figure. (**A**) Receiver operating characteristic (ROC) curves comparing the models developed in this research: a Gradient Boosting Decision Trees (GBDT) model of all features; logistic regression models of four blood tests; an anthropometry-based model compared to the well-established German Diabetes Risk Score (GDRS) and Finnish Diabetes Risk Score (FINDRISC). (**B**) Precision–recall (P-R) curves,

*Figure 3 continued*

showing the precision versus the recall for each model, with the prevalence of the population marked with the dashed line. (**C**) Deciles' odds ratio graph, the prevalence ratio in each decile to the prevalence in the fifth decile. (**D**) A feature importance graph of the logistic regression anthropometry model for a model with normalized features values. The bars indicate the feature importance values' standard deviation (SD). The top predictive features of this model are the body mass index (BMI) and waist-to-hip ratio (WHR). (**E**) Feature importance graph of logistic regression blood tests model with SD bars. While higher levels of HbA1c% positively contribute to type 2 diabetes (T2D) prediction, and high-density lipoprotein (HDL) cholesterol levels are negatively correlated with the predicted probability of T2D, the information provided by age and sex relevant for predicting T2D onset is screened by other features. (**F**) A calibration plot of the anthropometry, four blood tests, full blood test, and the FINDRISC models. Calibration of the models' predictions allows reporting the probability of developing T2D (see 'Methods').

The online version of this article includes the following source data for figure 3:

**Source data 1.** Detailed results for the top to bottom quantiles OR calculation.

**Source data 2.** Detailed coefficients for the non-laboratory logistic regression model.

**Source data 3.** Detailed coefficients for the laboratory logistic regression model.

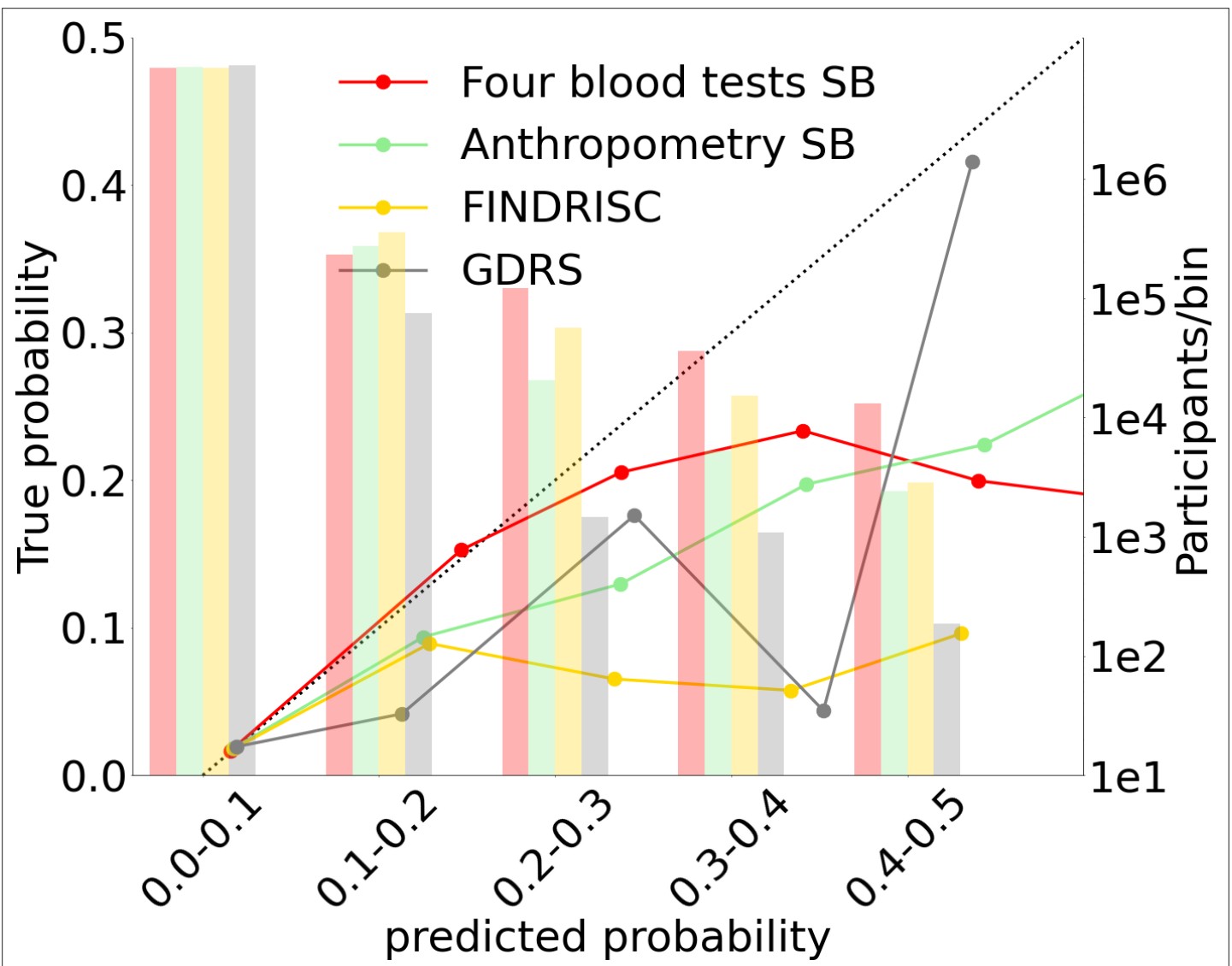

**Figure 4.** Models calibration plots. Anthropometric, four blood tests, Finnish Diabetes Risk Score (FINDRISC), and German Diabetes Risk Score (GDRS) scorecards calibration graphs.

**Table 2.** External validation cohort ('Clalit') statistical data.

| | Males (%) | HbA1c (%) | GGT | Reticulocyte count | HDL | Triglycerides | Age | Weight | Height | BMI |
|---|---|---|---|---|---|---|---|---|---|---|
| Number of samples | | 17,132 | 17,132 | 83 | 17,132 | 17,132 | 17,132 | 17,051 | 17,051 | 17,051 |
| Mean value | 45 | 5.56 | 32.31 | 56.33 | 49.77 | 141.33 | 56.40 | 79.00 | 1.66 | 28.72 |
| Standard deviation | | 0.41 | 49.29 | 36.97 | 13.33 | 82.09 | 8.06 | 49.90 | 0.09 | 19.58 |
| 0.25 | | 5.30 | 17.00 | 38.35 | 40.00 | 90.00 | 50.14 | 67.00 | 1.59 | 24.80 |
| 0.50 | | 5.60 | 23.00 | 58.00 | 48.00 | 123.00 | 57.02 | 77.00 | 1.65 | 27.68 |
| 0.75 | | 5.90 | 33.00 | 78.60 | 57.00 | 170.00 | 62.83 | 87.82 | 1.72 | 31.25 |

HbA1c, hemoglobin A1c; GGT, gamma-glutamyl transferase; HDL, high-density lipoprotein; BMI, body mass index.

criteria noted above. We then included data from the following tests: GGT (80,000 patients), HDL (151,969 patients), and triglycerides (157,321 patients). In addition to the HbA1c% exclusion criteria, we added the following: patients who did not have all four blood tests; patients older than 70 or younger than 40; patients who were diagnosed with T2D before the first visit; patients who had a diabetes diagnosis without a clear indication that it was T2D; and patients who had taken diabetes-related drugs (ATC code A10) before the first visit or before being diagnosed with T2D either based on HbA1c% levels or by a physician.

As a criterion for T2D, we considered two consecutive tests with HbA1c $\geq$ 6.5% or a physician diagnosis of T2D. After excluding patients according to the above criteria, the remaining cohort included 17,132 patients with anthropometric characteristics similar to the UKB cohort (**Table 2**). The remaining cohort's T2D onset prevalence is 4.1%, considerably greater than the 1.79% in the UKB cohort. We further tested the model on a stratified normoglycemic subcohort with 10,064 patients and a prevalence of 2% T2D and a prediabetes subcohort with 7059 patients with 7.1% T2D prevalence.

We tested the four blood test model on the data from the above cohorts by calculating a raw score for each participant based on all relevant scorecard features apart from the time 'years for prediction' feature. We then randomly sampled out of a normal distribution resembling the UKB cohort (mean = 7.3 years, SD = 2.3 years) 1000 time periods for a returning visit for each participant. We limited the patients' time of returning between 2 and 17 years to emulate the UKB data. The cutoff data for last visit was December 31, 2019, the last date reported in the Clalit database. We then estimated the mean and 95% CI of these cohort's auROC and APS results.

We did not evaluate the FINDRISC, GDRS, and anthropometrics models using the Clalit database as these models required some features that do not appear in the Clalit database. The FINDRISC model requires data regarding physical activity, waist circumference, and consumption of vegetables, fruit, or berries. The GDRS requires the following missing data fields: physical activity, waist circumference, consumption of whole-grain bread/rolls and muesli, consumption of meat, and coffee consumption. The anthropometrics model requires data regarding waist and hips circumference.

## Results

### Anthropometric-based model

Using the anthropometrics scorecard model the patient's final score relates to three risk groups (see 'Model building procedures'). Participants within the score range between 1 and 69 have a 1% (95% CI 0.8–1%) probability of developing T2D. The second group, with a score range between 70 and 78, predicts a 5% (95% CI 3–6%) probability of developing T2D. The third group, with a score range of between 79 and 96, predicts a 9% (95% CI 7–12%) probability of developing T2D (**Figure 2C**).

We also provide models with the same features in their LR form and a Cox regression form for more accurate computer-aided results. Testing these models using the holdout test set achieved an auROC of 0.81 (0.78–0.84) and an APS of 0.09 (0.06–0.13) at 95% CI. Applying a survival analysis Cox regression model to the same features resulted in comparable results (**Table 3**). Using the model in

**Table 3.** Comparing models' main results.

The values in parentheses indicate a 95% confidence interval (CI). The deciles' odds ratio (OR) measures the ratio between type 2 diabetes (T2D) prevalence in the top risk score decile bin and the prevalence in the fifth decile bin (see 'Methods').

| Measure type | Model type | APS | auROC | Decile's prevalence OR |
|---|---|---|---|---|
| GDRS | Score card cox regression for 5 years | 0.04 (0.03–0.06) | 0.66 (0.62–0.70) | 2.5 (1.46–4.45) |
| FINDRISC | Score card logistic regression | 0.04 (0.03–0.06) | 0.73 (0.69–0.76) | 4.13 (2.29–7.37) |
| Anthropometry | Score card cox regression for 5 years | 0.04 (0.03–0.07) | 0.79 (0.75–0.83) | 8.8 (3.6–36) |
| Anthropometry | Score card cox regression for 10 years | 0.06 (0.04–0.09) | 0.79 (0.76–0.82) | 10 (4.6–32.9) |
| Anthropometry | Score card logistic regression | 0.07 (0.05–0.10) | 0.81 (0.77–0.84) | 17.2 (5–66) |
| Anthropometry | Logistic regression | 0.09 (0.06–0.13) | 0.81 (0.78–0.84) | 16.9 (4.8–66) |
| Anthropometry | Cox regression | 0.10 (0.07–0.13) | 0.82 (0.79–0.85) | 10.7 (5–24) |
| Four blood tests | Score card cox regression for 10 years | 0.13 (0.10–0.16) | 0.87 (0.85–0.90) | 22.4 (9.8–54) |
| Four blood tests | LR score card | 0.13 (0.10–0.17) | 0.87 (0.85–0.90) | 48 (11.9–109) |
| Four blood tests | Score card cox regression for 5 years | 0.09 (0.06–0.12) | 0.89 (0.86–0.92) | 53.2 (18.9–84.2) |
| Four blood tests | Cox regression | 0.25 (0.18–0.32) | 0.88 (0.85–0.90) | 43 (13.6–109) |
| Four blood tests | Logistic regression | 0.24 (0.17–0.31) | 0.88 (0.85–0.91) | 32.5 (10.89–110) |
| Blood tests | Logistic regression | 0.26 (0.19–0.33) | 0.91 (0.89–0.93) | 75.4 (17.7–133) |
| All features | Boosting decision trees | 0.27 (0.20–0.34) | 0.91 (0.89–0.93) | 72.6 (15.1–135) |

APS, average precision score; auROC, area under the receiver operating curve; GDRS, German Diabetes Risk Score; FINDRISC, Finnish Diabetes Risk Score; DT, Decision Trees.

its scorecard form, we achieved an auROC of 0.81 (0.77–0.84) and an APS of 0.07 (0.05–0.10). All these models outperformed the two models that we used as a reference: applying the FINDRISC model resulting in an auROC of 0.73 (0.69–0.76) and an APS of 0.04 (0.03–0.06), and applying the GDRS model resulting in an auROC of 0.66 (0.62–0.70) and an APS of 0.04 (0.03–0.06) (*Figure 3A and B*, *Table 3*, and 'Methods'). With the cohort's baseline prevalence of 1.79%, the Cox regression model achieved deciles' odds ratio (OR) of ×10.65 (4.99–23.67), the L.R. Anthropometric model achieved deciles' OR of ×16.9 (4.84–65.93), and its scorecard derivative achieved deciles OR of ×17.15 (5–65.97) compared to the FINDRISC model's ×4.13 (2.29–7.37) and the ×2.53 (1.46–4.45) deciles' OR achieved by the GDRS model (*Figure 3C*, *Table 3*, 'Methods'). The WHR and BMI have the highest predictability in the anthropometric model (*Figure 3D*). These two body habitus measures are indicators associated with chronic illness (*Eckel et al., 2005*; *Cheng et al., 2010*; *Jafari-Koshki et al., 2016*; *Qiao and Nyamdorj, 2010*).

## Model based on four blood tests

Using the four blood tests scorecard ('Methods,' *Figure 2B*), we binned the resulting scores into four groups. Participants with a score within the score range of between 1 and 104 have a 0.5% (95% CI 0.4–0.7%) probability of developing T2D. The second group, with a score range of between 105 and 116, predicts a 3% (95% CI 2–4%) probability of developing T2D. The third group score, with a range of between 117 and 146, predicts a 10% (95% CI 8–12%) of developing T2D. The fourth group score range was between 147 and 162, and participants in this score range have a 23% (95% CI 10–37%) probability of developing T2D.

We used four common blood test scores as input to the survival analysis and the LR model. Applying the survival analysis Cox regression model for the test set, we achieved an ROC of 0.88 (0.85–0.90), an APS of 0.25 (0.18–0.32), and a deciles OR of ×42.9 (13.7–109.1). Using the four blood tests LR model, we achieved comparable results: an auROC of 0.88 (0.85–0.91), an APS of 0.24 (0.17–0.31), and a deciles' OR of 32.5 (10.8–110.1). Applying the scorecard model, we achieve an auROC of 0.87 (0.85–0.9), an APS of 0.13 (0.10–0.17), and a deciles' OR of 47.7 (79–115) (*Figure 3A–C*, *Table 3*). The

**Table 4.** Comparing model results applied to an HbA1c% stratified population.

The values in parentheses indicate 95% confidence interval (CI). Results of the models applied to a stratified population. The mixed population-based model column provides the results of the scorecard models presented in *Figure 2* applied to normoglycemic and prediabetes stratified population.

| Population | | Mixed population-based model: tested on a stratified population | | Models built using a stratified training set | |
|---|---|---|---|---|---|
| | | **auROC** | **APS** | **auROC** | **APS** |
| Prediabetic (N = 1006, prevalence = 9.4%) | GDRS | 0.64 (0.57–0.70) | 0.17 (0.12–0.23) | - | |
| | FINDRISC | 0.66 (0.61–0.72) | 0.20 (0.14–0.27) | - | |
| | Anthropometry | 0.73 (0.68–0.77) | 0.20 (0.15–0.26) | 0.73 (0.68–0.78) | 0.21 (0.16–0.27) |
| | Four blood tests | 0.73 (0.68–0.77) | 0.20 (0.15–0.26) | 0.72 (0.67–0.77) | 0.21 (0.15–0.26) |
| Normoglycemic (N = 7948, prevalence = 0.8%) | GDRS | 0.67 (0.61–0.74) | 0.02 (0.01–0.03) | - | |
| | FINDRISC | 0.74 (0.69–0.79) | 0.04 (0.02–0.07) | - | |
| | Anthropometry | 0.81 (0.76–0.86) | 0.04 (0.02–0.07) | 0.81 (0.76–0.85) | 0.03 (0.02–0.06) |
| | Four blood tests | 0.81 (0.76–0.85) | 0.03 (0.02–0.05) | 0.82 (0.77–0.86) | 0.05 (0.03–0.09) |

auROC, area under the receiver operating curve; FINDRISC, Finnish Diabetes Risk Score; GDRS, German Diabetes Risk Score; APS, average precision score.

four blood test model results are superior to those of the nonlaboratory anthropometric model and those of the commonly used FINDRISC and GDRS models (*Figure 3A–C*, *Table 3*).

As expected, the HbA1c% feature had the highest predictive power since it is one of the criteria for T2D diagnosis. The second-highest predictive feature was HDL cholesterol, which is known to be beneficial for health, especially in the context of cardiovascular diseases, with high levels being negatively correlated to T2D onset. (*Meijnikman et al., 2018*; *Kontush and Chapman, 2008*; *Bitzur et al., 2009*) . Interestingly, age and sex had a low OR value, meaning that they hardly contributed to the model, probably because of the T2D-relevant information of these features latent within the blood tests data.

We compared these results to those of 59 blood tests input features of the LR model and those of a GBDT model, including 13 features aggregations composed of 279 individual features and genetics data available in the dataset. These two models achieved an auROC of 0.91 (0.89–0.93) and 0.91 (0.9–0.93), an APS of 0.26 (0.19–0.33) and 0.27 (0.20–0.34), and a deciles' OR of ×75.4 (17.74–133.45) and ×72.6 (15.09–134.9), respectively.

## Prediction within an HbA1c% stratified population

To verify that our scorecard models can discriminate within a group of normoglycemic participants and within a group of prediabetic participants, we tested the models separately on each group. We separated the groups based on their HbA1c% levels during the first visit to the UKB assessment centers. We allocated participants with 4% ≤ HbA1c% < 5.7% to the normoglycemic group and participants with 5.7% = <HbA1c% < 6.5% levels to the prediabetic group (*Cheng et al., 2010*). As HbA1c% is one of the identifiers of T2D, this measure is a strong predictor of T2D. The prevalence of T2D onset within the normoglycemic group is only 0.8% versus a prevalence of 9.4% in the prediabetic group.

**Table 5.** Four blood tests scorecard results from the external validation cohort ('Clalit').

| Label | Cohort size | Prevalence (%) | APS | auROC |
|---|---|---|---|---|
| Full population (HbA1c% < 6.5%) | 17,132 | 4.1 | 0.11 (0.10–0.11) | 0.75 (0.74–0.75) |
| Normoglycemic population (HbA1c% < 5.7%) | 10,064 | 2 | 0.04 (0.04–0.05) | 0.69 (0.66–0.69) |
| Prediabetes population (5.7% = <HbA1c% < 6.5%) | 7059 | 7.1 | 0.12 (0.12–0.13) | 0.68 (0.67–0.69) |

APS, average precision score; auROC, area under the receiver operating curve.

The anthropometry model yielded an auROC of 0.81 (0.76–0.86) within the normoglycemic group with an APS of 0.04 (0.02–0.07). When testing the models within the prediabetic group, the anthropometry model achieved an auROC of 0.73 (0.68–0.77) and an APS of 0.2 (0.15–0.26). Both results outperform the FINDRISC and GDRS results. For the normoglycemic HbA1c% range, the four blood test model yielded an auROC of 0.81 (0.76–0.85) and an APS of 0.03 (0.02–0.05). These results are similar to those of the anthropometry model's results. To explore the option of developing scorecard models dedicated to these stratified populations, we developed and tested two such models, which achieved results similar to the mixed-cohort model (*Table 4*).

## Validating the four blood test model on an external independent cohort

To validate our four blood test model, we utilized the Israeli electronic medical record database of Clalit Health Services as an external cohort. Applying our model to nondiabetic participants of the same age range population (see 'Mmethods'), the four blood test model achieved an auROC of 0.75 (95% CI 0.74–0.75) and an APS of 0.11 (95% CI 0.10–0.11) on a population of 17,132 participants with a 4.1% T2D onset prevalence. We then tested the model on stratified normoglycemic and prediabetes subcohorts. In the normoglycemic population (N = 10,064 participants) with T2D onset prevalence of 2%, the model achieved an auROC of 0.69 (95% CI 0.66–0.69) and an APS of 0.04 (95% CI 0.04–0.05). Within the prediabetes population (N = 7059) with a T2D onset prevalence of 7.1%, the model achieved an auROC of 0.68 (95% CI 0.67–0.69) and an APS of 0.12 (95% CI 0.12–0.13) (*Table 5*). These results validate the general applicability of our models applied to an external cohort. As this database lacks data required for the anthropometry, GDRS, and FINDRISC scorecards, we could not apply these models to the Clalit database (see 'External validation cohort').

## Discussion and conclusions

In this study, we analyzed several models for predicting the onset of T2D, which we trained and tested on a UKB-based cohort aged 40–69. Due to their accessibility and high predictability, we suggest two simple scorecard models: the anthropometric and four blood test models. These models are suited for the UKB cohort or populations with similar characteristics (see *Table 1*).

To provide an accessible and simple yet predictive model, we based our first model on age, sex, and six nonlaboratory anthropometric measures. We then developed an additional, more accurate, straightforward model that can be used when laboratory blood test data are available. We based our second proposed model on four blood tests, in addition to age and sex of the participants. We reported results of both models according to their scoring on survival analysis Cox regression and LR models. As these models require computer-aided analysis, we developed an easy-to-use scorecard form. For all models, we obtained results that were superior to those of the current clinically validated nonlaboratory models, FINDRISC and GDRS. As a fair comparison, we trained these reference models and evaluated their predictions on the same datasets used with all our models.

Our models achieved a better auROC, APS, decile prevalence OR, and better-calibrated predictions than the FINDRISC and GDRS models. The anthropometrics and the four blood tests survival analysis models achieved deciles prevalence OR of ×10.7 and ×42.9, respectively, while the scorecard forms achieved OR of ×17.15 and ×47.7, respectively.

The anthropometry-based model retained its auROC performance of 0.81 (0.76–0.86) in the normoglycemic population but its performance worsened to 0.73 (0.68–0.77) in the prediabetes population. The four blood test model's performance showed a similar trend in these two subcohorts (*Table 4*). Training a subcohort-specific model did not improve these results.

Analyzing our models' feature importance, we conclude that the most predictive features of the anthropometry model are the WHR and BMI, body metrics that characterize body type or shape data. These features are known in the literature as being related to T2D, such conditions known as part of the metabolic syndrome (*Eckel et al., 2005*). The most predictive feature of the four blood test model is the HbA1c%, which is a measure of the glycated hemoglobin carried by the red blood cells, often used to diagnose diabetes. Interestingly, age and sex had a very low feature importance value, implying that they hardly contributed to the model results. One potential explanation is that the T2D-related information of these features is already latent within the blood test data. For instance, the sex

hormone-binding globulin (SHBG) feature contains a continuous measure regarding the sex hormone of each participant, thus making the sex feature redundant.

Applying the four blood test model to the Clalit external cohort, we achieved an auROC of 0.75 (0.74–0.75). While we obtained a sound prediction indication, the results are inferior to the scores when applied to the UKB population. This sound prediction indication and degradation in performance are seen both in the general population and in the HbA1C% stratified subcohorts. We expected degradation in results when transitioning from the UKB to the Clalit cohort as these two cohorts vary in many aspects. While the UKB is a UK population-based prospective study suffering from 'healthy volunteer selection bias' and from 'attrition bias' (*Fry et al., 2017*; *Hernán et al., 2004*), the Clalit cohort is a retrospective cohort based on an Israeli population and suffers from ascertainment bias and diagnostic suspicion biases, as people with higher risk for T2D are sent to perform the related blood tests. In both studies, there is a need to handle missing data. In the Clalit database, we had to drop patients with inconclusive diagnoses (e.g., diabetes diagnosis, without referring to the type of diabetes; see 'External validation cohort section'). One of the most apparent differences is seen when comparing the T2D prevalence of the two cohorts: 1.79% for the UKB versus 4.1% for the Clalit database.

One main limitation of our study is that our cohorts' T2D prevalence is biased away from the general UK populations' T2D prevalence. Our cohort's T2D prevalence was only 1.79%, while the general UK population's T2D prevalence is 6.3%, and 8% among adults aged 45–54 in the general UK population (2019) (Diabetes UK). This bias is commonly reported as a 'healthy volunteer' selection bias (*Fry et al., 2017*; *Hernán et al., 2004*), which reduces the T2D prevalence from 6% in the general UK population to 4.8% in the UKB population. An additional screening bias is caused by including only healthy participants on their first visit. This contributed to the reduced prevalence of T2D in our cohort of 1.79% T2D onset. Applying our models to additional populations requires further research, and fine-tuning of the feature coefficients might be required.

As several studies have concluded (*Knowler et al., 2002*; *Lindström et al., 2006*; *Diabetes Prevention Program Research Group, 2015*), a healthy lifestyle and diet modifications are expected to reduce the probability of T2D onset; therefore, identifying people at risk for T2D is crucial. We assert that our models make a significant contribution to such identification in two ways: the laboratory four blood test model for clinical use is highly predictive of T2D onset, and the anthropometrics model, mainly in its scorecard form, is an easily accessible and accurate tool. Thus, these models have the potential to improve millions of people's lives and reduce the economic burden on the medical system.

## Acknowledgements

This research has been conducted by using the UK Biobank Resource under Application Number 28784. We thank I Kalka, N Bar, MD, Yotam Reisner, MD, PhD, Smadar Shilo, PhD, E Barkan, and members of the Segal group for discussions.

## Additional information

### Funding

| Funder | Grant reference number | Author |
| --- | --- | --- |
| Feinberg Graduate School, Weizmann Institute of Science | | Eran Segal |

The funders had no role in study design, data collection and interpretation, or the decision to submit the work for publication.

### Author contributions

Yochai Edlitz, Conceptualization, Formal analysis, Investigation, Methodology, Software, Visualization, Writing - original draft; Eran Segal, Conceptualization, Methodology, Project administration, Supervision, Validation, Writing - review and editing

## Author ORCIDs
Yochai Edlitz ⬤ http://orcid.org/0000-0001-7733-3995
Eran Segal ⬤ http://orcid.org/0000-0002-6859-1164

## Decision letter and Author response
Decision letter https://doi.org/10.7554/eLife.71862.sa1
Author response https://doi.org/10.7554/eLife.71862.sa2

## Additional files

### Supplementary files
• Transparent reporting form

### Data availability
All data that we used to develop the models in this research is available through the UK Biobank database. The external validation cohort is from "Clalit healthcare". The two databases can be accessed upon specific requests and approval as described below. UKBiobank - The UK Biobank data is Available from UK Biobank subject to standard procedures (https://www.ukbiobank.ac.uk/enable-your-research/apply-for-access). The UK Biobank resource is open to all bona fide researchers at bona fide research institutes to conduct health-related research in the public interest. UK Biobank welcomes applications from academia and commercial institutes. Clalit - The data that support the findings of the external Clalit cohort originate from Clalit Health Services (http://clalitresearch.org/about-us/our-data/). Due to restrictions, these data can be accessed only by request to the authors and/or Clalit Health Services. Requests for access to all or parts of the Clalit datasets should be addressed to Clalit Healthcare Services via the Clalit Research Institute (http://clalitresearch.org/contact/). The Clalit Data Access committee will consider requests given the Clalit data-sharing policy. Source code for analysis is available at https://github.com/yochaiedlitz/T2DM_UKB_predictions, (copy archived at swh:1:rev:1e6b22e3d51d515eb065d7d5f46408f86f33d0b8).

The following previously published dataset was used:

| Author(s) | Year | Dataset title | Dataset URL | Database and Identifier |
|---|---|---|---|---|
| Bycroft C, Freeman C, Petkova D | 2018 | The UK Biobank resource with deep phenotyping and genomic data | http://biobank.ctsu.ox.ac.uk/crystal/label.cgi?id=100314 | biobank, 100314 |

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

# Appendix 1

## Exploring the full features' space using GBDT

To select model features, we analyzed the importance of features that we sought to relate to T2D. We analyzed the power of a predictive model with a vast amount of information and compared it to our minimal features models.

We started by sorting out a list of 279 preliminary features from the first visit to the UKB assessment center. In addition to these features, we used the UKB SNPs genotyping data and its calculated PRS.

We inspected the impact of various features groups using the lightGBM (*Ke et al., 2017*) gradient decision trees model using SHAP (*Lundberg et al., 2020*; *Lundberg and Lee, 2017*; *Appendix 1—figure 1*, see 'SHAP'). We aggregated the features into 13 separate groups: age and sex; genetics; early-life factors; socio-demographic; mental health; blood pressure and heart rate; family background and ethnicity; medication; diet; lifestyle and physical activity; physical health; anthropometry; and blood tests. All of these groups included age and sex features.

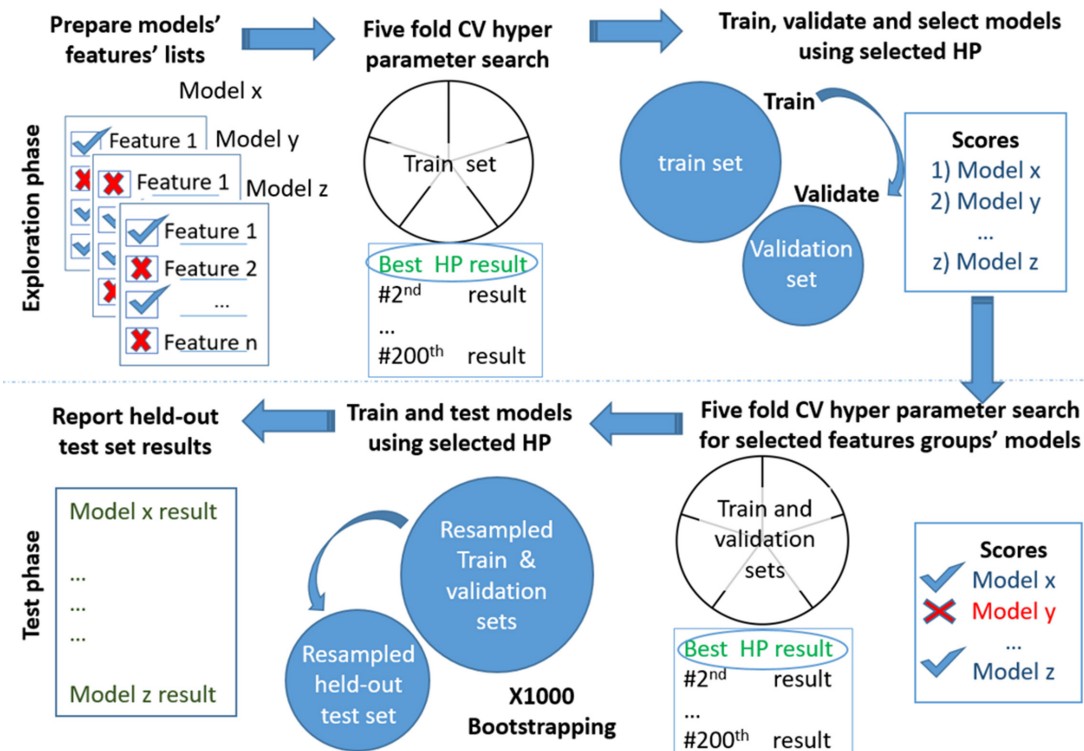

**Appendix 1—figure 1.** Models testing and training process. Models' development. Scheme of the models' exploration and evaluation process. For the models' selection process, we used a fivefold cross-validation with 200 iterations of the random hyperparameters process for each group of features. We then selected the top-scored hyperparameters for each feature's group. We trained a new model based on the training set and measured the area under the receiver operating curve (auROC) using the validation set. Out of the validated models, we chose the models that had a minimal number of features and provided high performance. The reported results are of the heldout test set.

We also tested the impact of HbA1c% with age and sex and genetics without age and sex. We list the top five predictive GBDT models, according to their auROC and APS, in descending order. 'All features without genetic sequencing' model – auROC of 0.92 (95% CI 0.89–0.94%) and APS of 0.28 (95% CI 0.20–0.36%). Adding genetics to this model degraded the results to an auROC of 0.91 (0.89–0.93) and an APS of 0.27 (0.2–0.34), probably due to overfitting. The 'full blood tests' model – auROC of 0.90 (0.88–0.93) and an APS of 0.28 (0.21–0.36). The 'four blood tests' model – auROC of 0.88 (0.85–0.90) and an APS of 0.20 (0.14–0.17). The HbA1c%-based model – auROC of 0.84 (0.80–0.87) and an APS of 0.17 (0.12–0.23). The anthropometry model – auROC of 0.79 (0.75–0.82) and an APS of 0.07 (0.05–0.11) (*Appendix 1—table 1*).

The lifestyle and physical activity model includes 98 features related to physical activity; addictions; alcohol, smoking, cannabis use, electronic device use; employment; sexual factors; sleeping; social support; and sun exposure. This model achieved an auROC of 0.73 (0.69–0.77), providing better prediction scores than the diet features group. The diet-based model includes 32 diet features from the UKB touchscreen questionnaire on the reported frequency of type and intake of common food and drink items. This model achieved an auROC of 0.66 (0.60–0.71).

**Appendix 1—table 1.** Predicting using feature domain groups.

Results of Gradient Boosting Decision Trees (GBDT) models for various feature domains.

| Label | APS | auROC |
|---|---|---|
| All features without genetic sequencing | 0.28 (0.20–0.36) | 0.92 (0.89–0.94) |
| All features | 0.27 (0.20–0.34) | 0.91 (0.89–0.93) |
| All blood tests | 0.28 (0.21–0.36) | 0.90 (0.88–0.93) |
| Four blood tests | 0.20 (0.14–0.27) | 0.88 (0.85–0.90) |
| Blood tests without HbA1c% | 0.13 (0.09–0.18) | 0.84 (0.81–0.87) |
| HbA1c% | 0.17 (0.12–0.23) | 0.84 (0.80–0.87) |
| Blood tests without HbA1c% nor glucose | 0.10 (0.07–0.13) | 0.82 (0.79–0.86) |
| Anthropometry | 0.07 (0.05–0.11) | 0.79 (0.75–0.82) |
| Lifestyle and physical activity | 0.05 (0.04–0.07) | 0.73 (0.69–0.77) |
| Blood pressure and heart rate | 0.05 (0.03–0.07) | 0.69 (0.64–0.73) |
| Nondiabetes-related medication | 0.04 (0.03–0.06) | 0.67 (0.62–0.73) |
| Mental health | 0.04 (0.03–0.06) | 0.67 (0.62–0.71) |
| Family and ethnicity | 0.04 (0.03–0.05) | 0.66 (0.60–0.71) |
| Diet | 0.04 (0.03–0.06) | 0.66 (0.60–0.71) |
| Socio-demographics | 0.03 (0.02–0.05) | 0.65 (0.60–0.70) |
| Early-life factors | 0.03 (0.02–0.05) | 0.64 (0.59–0.69) |
| Age and sex | 0.03 (0.02–0.04) | 0.61 (0.56–0.67) |
| Only genetics | 0.03 (0.02–0.04) | 0.57 (0.51–0.63) |

APS, average precision score; auROC, area under the receiver operating curve.

We then examined the additive contribution of each predictive group up to the total predictive power of the 'all features' model (*Appendix 1—table 2*). We started with the baseline model of 'age and sex' and added feature groups, sorted by their predictive power as separate GBDT models. We concluded that using the four blood test model substantially increases the accuracy of the prediction results compared to a model based only on HbA1c% and age and sex. The auROC and APS increases from 0.84 (0.80–0.87), 0.17 (0.12–0.23) to 0.88 (0.85–0.90), 0.20 (0.14–0.27), respectively.

The full blood test model increased the auROC and APS to 0.90 (0.88–0.93) and 0.28 (0.21–0.36), respectively. We did not identify any major increase in accuracy of the predictions by adding any other specific group to this list, suggesting that most of the predictive power of our models are either captured by the blood test features or possess collinear information. Using all features together provided an increase in performance to auROC, APS, and the deciles' OR of 0.9 (0.88–0.92), 0.28 (0.22–0.35), and ×65(49-73), respectively (*Appendix 1—table 2*).

**Appendix 1—table 2.** Summary of Incremental feature's model.

Comparison table of average precision score (APS) and area under the receiver operating curve (auROC) for the Gradient Boosting Decision Trees (GBDT) models, where each model includes the preceding model's features plus an additional feature domain. The largest increase in prediction accuracy was the result of adding the HbA1C% feature, which is also a biomarker for type 2 diabetes (T2D) diagnosis. Adding the DNA sequencing data did not significantly contribute to the prediction power of the model.

| Label | APS | auROC |
|---|---|---|
| Age and sex | 0.03 (0.02–0.04) | 0.61 (0.56–0.67) |
| HbA1c% | 0.17 (0.12–0.23) | 0.84 (0.80–0.87) |
| Four blood tests | 0.20 (0.14–0.27) | 0.88 (0.85–0.90) |
| All blood tests | 0.28 (0.21–0.36) | 0.90 (0.88–0.93) |
| Adding anthropometrics | 0.23 (0.17–0.30) | 0.90 (0.87–0.92) |
| Adding physical health DT | 0.28 (0.21–0.36) | 0.91 (0.89–0.93) |
| Adding lifestyle DT | 0.24 (0.18–0.32) | 0.91 (0.88–0.93) |
| Adding blood pressure and heart rate | 0.25 (0.19–0.33) | 0.91 (0.88–0.93) |
| Adding non-T2D-related medical diagnosis | 0.24 (0.18–0.32) | 0.91 (0.88–0.93) |
| Adding mental health | 0.28 (0.20–0.36) | 0.91 (0.89–0.93) |
| Adding medication | 0.28 (0.20–0.35) | 0.91 (0.89–0.93) |
| Adding diet | 0.24 (0.18–0.31) | 0.91 (0.89–0.93) |
| Adding family-related information | 0.28 (0.21–0.35) | 0.91 (0.89–0.94) |
| Adding early-life factors | 0.24 (0.17–0.31) | 0.91 (0.89–0.93) |
| Adding socio-demographic | 0.27 (0.20–0.36) | 0.92 (0.89–0.94) |
| Adding genetics | 0.27 (0.20–0.34) | 0.91 (0.89–0.93) |

## Deprivation index differences between sick and healthy populations in our UKB cohort

Here, we analyzed the Townsend deprivation index differences of participants diagnosed with T2D in one of their returning visits to the assessment center versus the healthy population. The Townsend deprivation index measures deprivation based on employment status, ownership of car and home, and overcrowded household. Higher index values represent lower socioeconomic status. According to our analysis, a higher deprivation index is correlated with a higher risk of developing T2D (*Appendix 1—figure 2A*). We analyzed the data using a Mann–Whitney $U$-test with a sample of 1000 participants from each group; we achieved a p-value of $2.37e^{-137}$. When we analyzed the full cohort, the p-value dropped below the computational threshold, indicating a significant correlation between a Townsend deprivation index and our cohort's tendency to develop T2D.

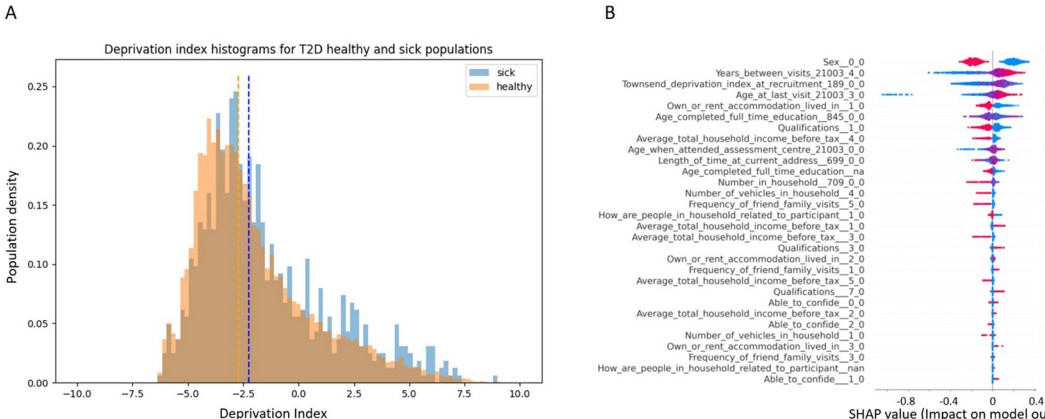

**Appendix 1—figure 2.** Socioeconomic impact on prediction of risk of developing type 2 diabetes (T2D). (**A**) Deprivation index differences between T2D sick and healthy populations in our data: a density histogram showing the differences in deprivation index of participants who were diagnosed with T2D in one of their returning visits to the assessment center and for healthy participants. Executing a Mann–Whitney test on this data yields a p-value lower than $2.37 * 10^{-137}$, indicating a correlation between lower socio-demographic state with higher T2D prevalence. (**B**) Shapley Additive Explanations (SHAP) analysis of the socio-demographic features for a Gradient

*Appendix 1—figure 2 continued*

Boosting Decision Trees (GBDT) predictor of T2D: Each dot represents a participant's value for each feature along the Y-axis. The colors indicate the values of the features: red indicates higher feature values, blue indicates lower feature values. The X-axis is the SHAP value, where higher SHAP values indicate a stronger positive impact on the positive prediction of the GBDT predictor, that is, higher risk for T2D onset. The analysis indicates that higher values of deprivation index and lower household income push the probability of T2D onset to higher values. The full meaning of the codes is provided at the UK Biobank data showcase.

We performed a SHAP analysis on the socio-demographic features of the patients, and features such as higher Townsend deprivation index or being in the lower-income groups (<40,000 GBP) push towards a prediction of developing T2D while being in the top two groups (52,000 GBP or more) is predictive of having less risk of developing T2D (Figure S3B). The full meaning of the codes is available at the UK Biobank data showcase.

## References for PRS summary statistics articles

HbA1c *Soranzo et al., 2010*; *Walford et al., 2016*; *Wheeler et al., 2017*; cigarettes per day, ever smoked, age start smoking *Tobacco and Genetics Consortium, 2010*; HOMA-IR, HOMA-B, diabetes BMI unadjusted, diabetes BMI adjusted, fasting glucose *Morris et al., 2013*; fasting glucose, 2 hr glucose level, fasting insulin, fasting insulin adjusted BMI'- (MAGIC Scott) *Scott et al., 2012*; fasting glucose, fasting glucose adjusted for BMI, fasting insulin adjusted for BMI *Manning et al., 2012*; Two hours glucose level *Saxena et al., 2010*; Fasting insulin *the DIAbetes Genetics Replication And Meta-analysis (DIAGRAM) Consortium, 2012*; Fasting Proinsulin *Strawbridge et al., 2011*; Leptin adjusted for BMI *Kilpeläinen et al., 2016*; Leptin unadjusted for BMI; Triglycerides, Cholesterol, LDL, hdl *Willer et al., 2013*; BMI *Locke et al., 2015*; Obesity class1, obesity_class2, overweight *Berndt et al., 2013*; Anorexia *Boraska et al., 2014*; Height *Wood et al., 2014*; Waist circumference, hips circumference *Shungin et al., 2015*; Cardio *Deloukas et al., 2013*; Heart_Rate *den Hoed et al., 2013*; Alzheimer *Lambert et al., 2013*; Asthma *Moffatt et al., 2010*.

