## [Editor Report]

The authors have used the UK Biobank with sophisticated statistical modeling to predict the risk of type 2 diabetes mellitus development. Prognosis and early detection of diabetes are key factors in clinical practice, and the current data suggest a new machine-learning-based algorithm that further advances our ability to prevent diabetes.

---

## [Decision Letter]

**Decision letter after peer review:**

Thank you for submitting your article "Prediction of type 2 diabetes mellitus onset using logistic regression-based scorecards" for consideration by *eLife*. Your article has been reviewed by 3 peer reviewers, and the evaluation has been overseen by a Reviewing Editor and a Senior Editor. The following individuals involved in review of your submission have agreed to reveal their identity: Sina Azadnajafabad (Reviewer #1); Promi Das (Reviewer #3).

Essential revisions:

1) Methodological issues and subjects inclusion raised in points 2-5 of reviewer 2 are crucial for the potential acceptance of this manuscript.

2) Further validation with another cohort as requested by reviewer 3 is needed.

*Reviewer #1 (Recommendations for the authors):*

1. Abstract, background: authors claim that their aim was to propose non-lab-based models for the use of lower socio-economic countries. However, almost half of the methods of this paper are on lab-based models. A revision of the aim of the study is necessary.

2. Abstract, background: "Early detection of T2D high-risk patients can reduce the incidence of the disease through a change in lifestyle, diet, or medication." Incidence of a disease is a multi-dimensional phenomenon and the claim that early detection of high-risk patients could reduce the incidence of a disease is not clinically sound. Maybe changing the sentence to a condition that this early detection may provide health authorities the proper vision to prepare the health systems for upcoming events be a better idea.

3. Abstract, methods: the "scoreboard form" and the comparison of the developed models with two previous prediction models should be explained in the methods clearly before proposing results and conclusion.

4. Introduction: the first part on the epidemiology of diabetes needs more updated statistics and references. There are multiple databases like the updated Global Burden of Disease 2019 database that authors could use. Also, comparing the burden of diabetes in various socio-economic levels of countries could benefit this section.

5. Methods: this section needs a clear elaboration on the reason for choosing the mentioned variables for non-lab and lab models. Definitely, the statistical aspects are well drafted. However, the manuscript needs a simple explanation for this issue.

6. Results: well-drafted and visualized.

7. Discussion and conclusion: a major part is missing on the link of the utilization of these models and reducing the burden of diabetes. Whether individual or population investigation and implementation of such models would be better needs to be discussed, providing essential points for those who want to benefit from what was introduced in this study.

*Reviewer #2 (Recommendations for the authors):*

Here is a summary of my main concerns:

1. The authors don't mention previous work of predicting T2D, except the GDRS and FINDRISC. There are many such studies, including studies that use the UKB. To name a few: Di Camillo et al., European Journal of Endocrinology, 2018; Lama et al., Heliyon, 2021; Zhang et al., Scientific Reports, 2020; Dolezalova et al., arxiv, 2021; He et al., Diabetes Care, 2021. This is just a few found on simple google search, but there are many more.

2. The numbers from the UKB don't look right to me. There is available clinical data for UKB participants, therefore no need to focus only on those that came back for additional visit. This limits the data in the study to ~70K individuals, and after exclusions to ~45K with only ~1K cases. In comparison, a recent study (He et al., Diabetes Care 2021) that used the same dataset, there were 7513 T2D cases.

3. It is also unclear to me how the "years of prediction" is calculated – is that the time between first and second visit? If so, that doesn't represent the time between the first visit and the in identification of T2D. This might be a major issue that needs to be addressed.

4. In addition, in this study a simple logistic regression method was used. However, this is a clear case of censored data. LR is not the right method for these kind of prediction tasks.

5. Another issue I find with the data is that pre-diabetic individuals are not excluded. Predicting that someone that is pre-diabetic will be diabetic is a very different task than predicting healthy individual to become diabetic. I understand that in the first model, we assume that blood tests are unavailable, so in theory in this model a pre-diabetic individual will not have access to HbA1c test and won't know that they are pre-diabetic. However, in the UKB cohort, that person knows about the condition, and thus, it confounds the prediction. I do see that a model without the pre-diabetic individuals was performed, but it is only a secondary analysis, and I think it should be the main analysis.

6. The issues above make it hard to compare the results in this study with previous studies. Previous analyses (including using the UKB) with GDRS and FINDRISC have showed an AUC of about 0.75. I find it hard to believe the GDRS results are only 0.58. This suggests that there are inconsistencies in the data and analysis in this study.

*Reviewer #3 (Recommendations for the authors):*

Specific suggestion:

As the model metrics and the cohort chosen are very similar to one another, it is highly suggested to conduct such analyses on a different country cohort if possible, the findings would be of additional value.

[Editors’ note: further revisions were suggested prior to acceptance, as described below.]

Thank you for resubmitting your work entitled "Prediction of type 2 diabetes mellitus onset using logistic regression-based scorecards" for further consideration by *eLife*. Your revised article has been evaluated by Matthias Barton (Senior Editor), a Reviewing Editor, and the original reviewers.

The manuscript has been improved but there are some remaining issues that need to be addressed, as outlined below:

*Reviewer #1 (Recommendations for the authors):*

The authors of this study made their best to address the comments and improve the draft in this revision. Although the changes based on my previously provided comments are enough and sound, the manuscript is a little bit messy and needs a comprehensive language revision and checking. For example, changing the sort of the sections of the manuscript has caused some errors in the number of sections and subsections. A general revision in this regard could finalize the manuscript in my opinion.

*Reviewer #2 (Recommendations for the authors):*

The authors fixed most of my concerns, but I still have some unresolved issues.

1. In the response the authors explain that they don't want to use the full cohort of patients with baseline information alone. The logic used is that there is 'diagnostic access' bias. It might be true, but its unclear to me how much this is a concern in UK. If this was a major concern, then it affects the questionnaires, not just the clinical data. It should also be noted that there is 'report' bias (as the authors note in the methods). I also disagree that those individuals that returned to another visit can be regarded as a "controlled cohort", on the contrary, this is a selected group, not randomized, as in the first visit. Finally, as I noted in the previous review, the 'time to event' is wrong – its not ~7 years to diagnosis – its ~7 years between visits. The outcome is an answer whether the individual has T2D or not, and it could have manifested 6 years earlier. For all these reasons and more (cohort size, selection bias, etc.) I urge you to reconsider, use the full cohort and infer outcome of T2D from the clinical data, not the questionnaire.

2. This continues to my previous concerns – I was happy to see the addition of a Cox model, but I can't understand why the logistic regression model is still being used. If you insist of using such a model, please don't refer to it as prediction of T2D ~7 years in advance – it's a prediction of answering true to a question whether you were diagnosed with T2D in the time between first and second visit. All the models, including the scorecards, should be based on the 'real' time to diagnosis. If time-varying models do not fit with a scorecard, you can create a model that predicts diabetes 1,2 or 5 years in advance.

3. Another comment I had that is still an issue is regarding the deciles fold-ratio. Confidence intervals are good, but as I noted, the ratio should not be between the top and bottom deciles but top and median deciles. The current approach can provide very impressive results for a useless model (for example – ~0% in bottom decile, ~1% in all other deciles).

4. It is great to see the external validation cohort. It would have been great to see the other models implemented in that external cohort (GDRC and FINDRISC) and get some sense how much this model can improve current risk stratification approaches.

5. The paper still requires major editing and grammar corrections.

*Reviewer #3 (Recommendations for the authors):*

Each of my suggestions has been sufficiently addressed and has been added to the updated manuscript by the authors.

---

## [Author Response]

Reviewer #1 (Recommendations for the authors):1. Abstract, background: authors claim that their aim was to propose non-lab-based models for the use of lower socio-economic countries. However, almost half of the methods of this paper are on lab-based models. A revision of the aim of the study is necessary.

We thank the reviewer for providing this comment, we changed the Background section of the Abstract to describe better the paper's objective and the models that we developed, which is the need for accurate yet accessible prediction models.

Abstract:

“Since populations of lower socio-demographic status are more susceptible to T2D and might have limited resources or access to a computer for prediction methods – there is a need for accessible yet accurate prediction models.”

Introduction: “where laboratory diagnostic testing and computer-based models might be limited for populations in these countries^”^

2. Abstract, background: "Early detection of T2D high-risk patients can reduce the incidence of the disease through a change in lifestyle, diet, or medication." Incidence of a disease is a multi-dimensional phenomenon and the claim that early detection of high-risk patients could reduce the incidence of a disease is not clinically sound. Maybe changing the sentence to a condition that this early detection may provide health authorities the proper vision to prepare the health systems for upcoming events be a better idea.

We understand the reviewer’s concerns that incidence of a disease is a multi-dimensional phenomenon, as such we toned down the statement to:

“Early detection of T2D high-risk patients may help to delay or reduce the incidence of the disease through a change in lifestyle, diet, or medication”

We would like to emphasize that we base our statement on several well-established peer reviewed papers which we cite here and in the paper in *line 58*1–3 , we also follow the WHO claims “A healthy diet, regular physical activity, maintaining a normal body weight and avoiding tobacco use are ways to prevent or delay the onset of type 2 diabetes” 4.

The International Diabetes federation claims that “While there are a number of factors that influence the development of type 2 diabetes, it is evident that the most influential are lifestyle behaviours commonly associated with urbanization. These include consumption of unhealthy foods and inactive lifestyles with sedentary behaviour. Randomised controlled trials from different parts of the world, including Finland, USA, China and India, have established the that lifestyle modification with physical activity and/or healthy diet can delay or prevent the onset of type 2 diabetes.”(International Diabetes Federation n.d.)

3. Abstract, methods: the "scoreboard form" and the comparison of the developed models with two previous prediction models should be explained in the methods clearly before proposing results and conclusion.

We thank you for this comment, we added to the Abstract, methods the following paragraph to better explain the GDRS and FINDRISC models:

“Using the scorecard models, a patient can easily mark its result in each scorecard question. The patient then sums up its final score indicating the related risk group for developing T2D. We compare our results to the Finnish Diabetes Risk Score (FINDRISC) and the German Diabetes Risk Score (GDRS). The FINDRISC is a commonly used, non-invasive T2D risk-score model, estimates patients aged between 35 and 64 risks of developing T2D within the next ten years based on a Finnish population. The German Diabetes Risk Score (GDRS) estimates a five-year risk for developing T2D. The GDRS is based on a population aged 35-65 years from the European Prospective Investigation into Cancer and Nutrition (EPIC)-Potsdam study. Both models are available as a scorecard form.”

4. Introduction: the first part on the epidemiology of diabetes needs more updated statistics and references. There are multiple databases like the updated Global Burden of Disease 2019 database that authors could use. Also, comparing the burden of diabetes in various socio-economic levels of countries could benefit this section.

We agree with the reviewer input, and we updated the statistics and reference as well as adding data regarding the comparison between high- and low-income countries.

5. Methods: this section needs a clear elaboration on the reason for choosing the mentioned variables for non-lab and lab models. Definitely, the statistical aspects are well drafted. However, the manuscript needs a simple explanation for this issue.

We accept this remark, and we updated section 4.2 in the Methods section to elaborate more about the reason for choosing the mentioned variables for non-lab and lab models.

6. Results: well-drafted and visualized.7. Discussion and conclusion: a major part is missing on the link of the utilization of these models and reducing the burden of diabetes. Whether individual or population investigation and implementation of such models would be better needs to be discussed, providing essential points for those who want to benefit from what was introduced in this study.

We accept the remark, and we edited the discussion and conclusion section to include:

“To apply our models to additional populations, further research on their ethnicity and fine-tuning of the feature coefficients might be required. Such fine tuning can be achieved by retraining models based on the features selected here on a representative cohort of the population under consideration.

As several studies have concluded 7^,^8^,^9, a healthy lifestyle and diet modifications before the inception of T2D are expected to reduce the probability of T2D onset. Therefore, identifying people at risk for this disease is crucial. We assert that our models make a significant contribution to such identification in two ways: The laboratory four blood tests model for clinical use is highly predictive of T2D onset, and the anthropometrics mode, in its scorecard form, is an easily accessible and accurate tool. Once an individual is identified as being at an elevated risk for T2D, recommendations for lifestyle and diet change should be recommended by a physician or a dietitian to reduce the risk of T2D onset. We recommend monitoring blood glucose levels or HbA1c% levels to reduce the risk of developing aT2D without being diagnosed and reduce the potentially severe damage to health of untreated T2D patients. Such models can also be carried out as survey tools to provide health authorities forecasts for future T2D onset in the population and thus take the appropriate actions at a population health level. Thus, these models carry the potential to improve millions of people's lives and reduce the economic burden on the medical system.”

Reviewer #2 (Recommendations for the authors):Here is a summary of my main concerns:1. The authors don't mention previous work of predicting T2D, except the GDRS and FINDRISC. There are many such studies, including studies that use the UKB. To name a few: Di Camillo et al., European Journal of Endocrinology, 2018; Lama et al., Heliyon, 2021; Zhang et al., Scientific Reports, 2020; Dolezalova et al., arxiv, 2021; He et al., Diabetes Care, 2021. This is just a few found on simple google search, but there are many more.

We thank the reviewer for providing these useful and constructive comments and for sharing his concerns. Following this note, we now added and discuss some models from recent years including invasive models that are more relevant for the Four blood tests model. The reason we choose to focus on the GDRS and FINDRISC models as a base for comparison is that these models are broadly used as scorecards and can be easily compared as apples to apples as an accessible and easy-to-use prediction model. Additional benefit of comparing our model to the GDRS and FINDRISC models is that these models are commonly used as a benchmark for other models (Di Camillo et al., 2018)(Lama et al., 2021)(Zhang et al., 2020)

2. The numbers from the UKB don't look right to me. There is available clinical data for UKB participants, therefore no need to focus only on those that came back for additional visit. This limits the data in the study to ~70K individuals, and after exclusions to ~45K with only ~1K cases. In comparison, a recent study (He et al., Diabetes Care 2021) that used the same dataset, there were 7513 T2D cases.

We thank the reviewer for this comment. Indeed, working with larger data sets may supply higher confidence in the results. Although we see the advantage of working with larger data amount, there are some disadvantages such as noisy and incomplete data compared to data that is collected in the controlled environment of the assessment center, which we used here. Some biases of data collected from electronic health records may include access to health care, language barriers, or other socioeconomic factors. Patients from low socioeconomic status may suffer from “Diagnostic access” bias and be might approach teaching clinics, where documentation or clinical reasoning may be less accurate or systematically different than the care provided to patients of higher socioeconomic status.(Gianfrancesco et al., 2018) Other biases may include ascertainment bias and diagnostic suspicion biases (Banerjee A, Pluddemann A, O’Sullivan J n.d.). As such, while we understand the advantages of having larger datasets, in our view the benefits of working with controlled datasets overcome the potential data shifts and biases despite the compromise on the number of participants in the cohort and the built in healthy bias cohort of the UK Biobank prospective dataset.

3. It is also unclear to me how the "years of prediction" is calculated – is that the time between first and second visit? If so, that doesn't represent the time between the first visit and the in identification of T2D. This might be a major issue that needs to be addressed.

We thank the reviewer for this insight. We now added a survival analysis where we use the UK biobank data field id 2976- “age diabetes diagnosed” as the time to the diagnosis. We show that the Survival Analysis results achieve comparable results to the logistic regression results that we computed before.

**Author response table 1. sa2table1:** Comparing models main results.

Label	Model type	APS	auROC	Deciles prevalence odds ratio
GDRS SA	Scoreboard	0.04 (0.03-0.06)	0.66 (0.62-0.70)	11 (3.8-38)
FINDRISC LR	Scoreboard	0.04 (0.03-0.06)	0.73 (0.69-0.76)	33(9.6-67)
Anthropometry	Scoreboard	0.07(0.05-0.10)	0.81(0.77-0.84)	54(18-79)
Anthropometry	Logistic regression	0.09(0.06-0.13)	0.82(0.78-0.84)	54(18-80)
Anthropometry	Cox regression	0.10(0.07-0.13)	0.82(0.79-0.85)	69(27-89)
Four blood tests	Scoreboard	0.13(0.10-0.17)	0.87(0.85-0.90)	96(79-115)
Four blood tests	Cox regression	0.25(0.18-0.32)	0.88(0.85-0.90)	101(84-121)
Four blood tests	Logistic regression	0.24(0.17-0.31)	0.88(0.85-0.91)	104(84-125)
Blood tests	Logistic regression	0.26(0.19-0.33)	0.91(0.89-0.93)	116(95-138)
All features DT	Boosting decision trees	0.27(0.20-0.34)	0.91(0.89-0.93)	117(98-139)

For the logistic regression models, we indeed used the UK biobank data field “2443-Diabetes diagnosed by doctor” at the last visit as a time-point where we predict the status of the patient.

In addition, in this study a simple logistic regression method was used. However, this is a clear case of censored data. LR is not the right method for these kind of prediction tasks.

We agree that SA is a suitable choice for such data and we added survival analysis to the paper. We use the UK biobank data field id 2976- “age diabetes diagnosed” as the time to the diagnosis.

Another issue I find with the data is that pre-diabetic individuals are not excluded. Predicting that someone that is pre-diabetic will be diabetic is a very different task than predicting healthy individual to become diabetic. I understand that in the first model, we assume that blood tests are unavailable, so in theory in this model a pre-diabetic individual will not have access to HbA1c test and won't know that they are pre-diabetic. However, in the UKB cohort, that person knows about the condition, and thus, it confounds the prediction. I do see that a model without the pre-diabetic individuals was performed, but it is only a secondary analysis, and I think it should be the main analysis.

We thank the reviewer for this comment, and we also had this deliberation arising from the same concerns that the reviewer raised. To improve that point, we took the following actions:

1. We fitted scoreboard models for the prediabetes and the normoglycemic subpopulations for both the Anthropometry model and for the Four blood tests model.

2. We compared these models results to the results of the model of the entire population and we show that the two models supply comparable results (2.3 Prediction within an HbA1c% stratified population) and Table 3 (Comparing models results on HbA1c% stratified population). We show that the differences between the performance of the entire population models Vs. The stratified population models are not large. We thus believe that using a unified model for the two sub cohorts is more useable and accessible and can reduce confusion of potential users.

3. We added the analysis of the Pre-diabetes and the normoglycemic sub-cohorts to the paper’s main part (Section 2.3), and we show that the anthropometry model achieves results similar to the four blood tests model in the normoglycemic sub cohort.

6. The issues above make it hard to compare the results in this study with previous studies. Previous analyses (including using the UKB) with GDRS and FINDRISC have showed an AUC of about 0.75. I find it hard to believe the GDRS results are only 0.58. This suggests that there are inconsistencies in the data and analysis in this study.

We thank the reviewer for highlighting this point, we investigated this issue and indeed found an error in the analysis of the GDRS model. The new results that the GDRS model achieves are auROC of 0.66 (0.62-0.70) and an APS (Average Precision Score) of 0.04 (0.03-0.06). The new results still show an inferior result compared to the other models that we tested.

Reviewer #3 (Recommendations for the authors):Specific suggestion:As the model metrics and the cohort chosen are very similar to one another, it is highly suggested to conduct such analyses on a different country cohort if possible, the findings would be of additional value.

We thank the reviewer for this comment. Following this comment, we exploited the Clalit retrospective cohort’s electronic health records as an external validation cohort for the four blood tests scorecard model. Clalit is the largest Israeli healthcare organization, serving more than 4.4 million patients. Clalit database holds electronic health records of over 11 million patients, dating back to 2002, and is considered as one of the world’s largest EHR database. As our anthropometry models require hips and waist circumference as features for the model, which are not measure by the Clalit, we couldn't test this model as well on the Clalit dataset. An additional outcome of this validation procedure, was to find that “Reticulocytes” is an exceedingly rare blood test. To keep the accessibility of the model, we dropped the Reticulocytes count from the model, and rerun all analysis without this feature.

References

1. Knowler, W. C. et al. Reduction in the incidence of type 2 diabetes with lifestyle intervention or metformin. N. Engl. J. Med. 346, 393–403 (2002).

2. Diabetes Prevention Program Research Group et al. 10-year follow-up of diabetes incidence and weight loss in the Diabetes Prevention Program Outcomes Study. Lancet 374, 1677–1686 (2009).

3. Diabetes Prevention Program Research Group. Long-term effects of lifestyle intervention or metformin on diabetes development and microvascular complications over 15-year follow-up: the Diabetes Prevention Program Outcomes Study. Lancet Diabetes Endocrinol. 3, 866–875 (2015).

4. WHO | Diabetes programme. https://web.archive.org/web/20140329084830/http://www.who.int/diabetes/en/.

[Editors’ note: further revisions were suggested prior to acceptance, as described below.]

Reviewer #1 (Recommendations for the authors):The authors of this study made their best to address the comments and improve the draft in this revision. Although the changes based on my previously provided comments are enough and sound, the manuscript is a little bit messy and needs a comprehensive language revision and checking. For example, changing the sort of the sections of the manuscript has caused some errors in the number of sections and subsections. A general revision in this regard could finalize the manuscript in my opinion.

We thank the reviewer for this input. We revised the paper and we edited the article accordingly. We rearranged the structure of the article and improved the grammar and order.

Reviewer #2 (Recommendations for the authors):The authors fixed most of my concerns, but I still have some unresolved issues.1. In the response the authors explain that they don't want to use the full cohort of patients with baseline information alone. The logic used is that there is 'diagnostic access' bias. It might be true, but its unclear to me how much this is a concern in UK. If this was a major concern, then it affects the questionnaires, not just the clinical data. It should also be noted that there is 'report' bias (as the authors note in the methods). I also disagree that those individuals that returned to another visit can be regarded as a "controlled cohort", on the contrary, this is a selected group, not randomized, as in the first visit. Finally, as I noted in the previous review, the 'time to event' is wrong – its not ~7 years to diagnosis – its ~7 years between visits. The outcome is an answer whether the individual has T2D or not, and it could have manifested 6 years earlier. For all these reasons and more (cohort size, selection bias, etc.) I urge you to reconsider, use the full cohort and infer outcome of T2D from the clinical data, not the questionnaire.

We thank and appreciate this input, but unfortunately, we do not have data from the UK Biobank that includes the first diagnosis of T2D, and only have the date of T2D onset for the returning participants.

The biases in the returning visits are somewhat reduced due to the fact that UKBB invites returning participants randomly. We thus believe that despite the compromises on the number of participants in the cohort – this cohort holds some benefits over working with data from electronic health records (EHR), since EHR data is often incomplete compared to data collected in a controlled environment of an assessment center such as that which we use here.

Regarding the prediction of T2D onset, we corrected "years to diagnosis" to imply "years between visits."

2. This continues to my previous concerns – I was happy to see the addition of a Cox model, but I can't understand why the logistic regression model is still being used. If you insist of using such a model, please don't refer to it as prediction of T2D ~7 years in advance – it's a prediction of answering true to a question whether you were diagnosed with T2D in the time between first and second visit. All the models, including the scorecards, should be based on the 'real' time to diagnosis. If time-varying models do not fit with a scorecard, you can create a model that predicts diabetes 1,2 or 5 years in advance.

We thank the reviewer for this suggestion, and we understand the logic of applying survival analysis for such models.

As such, and following this comment, we developed a scorecard that relies on a Cox regression model for a fixed time point. In the comparison of Survival analysis to logistic regression results on our data, we found no major differences in the objective outcomes, as can be seen in Author response table 2.

**Author response table 2. sa2table2:** 

Label	Model type	APS	auROC	Decile’s prevalence odds ratio
Anthropometry	SA Scoreboard 5yrs	0.04 (0.03-0.07)	0.79 (0.75-0.83)	8.8 (3.6-36)
Anthropometry	SA Scoreboard 10yrs	0.06 (0.04-0.09)	0.79 (0.76-0.82)	10 (4.6-32.9)
Anthropometry	Scoreboard	0.07 (0.05-0.10)	0.81 (0.77-0.84)	17.2 (5-66)
Anthropometry	Logistic regression	0.09 (0.06-0.13)	0.81 (0.78-0.84)	16.9 (4.8-66)
Anthropometry	Cox regression	0.10 (0.07-0.13)	0.82 (0.79-0.85)	10.7 (5-24)
Four blood tests	SA Scoreboard 10yrs	0.13 (0.10-0.16)	0.87 (0.85-0.90)	22.4 (9.8-54)
Four blood tests	Scoreboard	0.13 (0.10-0.17)	0.87 (0.85-0.90)	48 (11.9-109)
Four blood tests	SA Scoreboard 5yrs	0.09 (0.06-0.12)	0.89 (0.86-0.92)	53.2 (18.9-84.2)
Four blood tests	Logistic regression	0.24 (0.17-0.31)	0.88 (0.85-0.91)	32.5 (10.89-110)
Four blood tests	Cox regression	0.25 (0.18-0.32)	0.88 (0.85-0.90)	43 (13.6-109)

Moreover, a survival analysis of our model will force a fixed timeframe for a potential T2D onset timeframe. We also point out that logistic regression is part of the common scorecards development pipeline. For example, the most common scorecard in use today, the FINRISC, is based on logistic regression. As such, we believe that showing the similarity in results of the SA analysis model to the LR model, while keeping the LR model-based scorecards provides for a flexible and easy to use score cards for assessing T2D.

3. Another comment I had that is still an issue is regarding the deciles fold-ratio. Confidence intervals are good, but as I noted, the ratio should not be between the top and bottom deciles but top and median deciles. The current approach can provide very impressive results for a useless model (for example – ~0% in bottom decile, ~1% in all other deciles).

We understand this concern, and we updated the deciles odds ratio to be relative to the fifth decile instead of the first decile. Author response table 2 and table 3 in the paper.

4. It is great to see the external validation cohort. It would have been great to see the other models implemented in that external cohort (GDRC and FINDRISC) and get some sense how much this model can improve current risk stratification approaches.

We agree that seeing the other models implemented in the external cohort (GDRS and FINDRISC) and getting some sense of how much our model can improve current risk stratification approaches would be valuable. We thus attempted to do so, but unfortunately, these models require some features that do not appear in the Clalit database. The FINRISC model requires data regarding physical activity, waist circumference, and consumption of vegetables, fruit, or berries. The GDRS requires data fields that are also missing: Physical activity; waist circumference, consumption of whole-grain bread/rolls and muesli; consumption of meat; Coffee consumption; And the Anthropometrics model requires data which we are missing regarding waist and hips circumference.

5. The paper still requires major editing and grammar corrections.

We thank the reviewer for this input. We revised the paper and we edited the article accordingly. We rearranged the structure of the article and improved the grammar and order.

Reviewer #3 (Recommendations for the authors):Each of my suggestions has been sufficiently addressed and has been added to the updated manuscript by the authors.

We thank the reviewer for the remarks and for improving our paper.